# Towards a Theoretical Understanding of Synthetic Data in LLM Post-Training: A Reverse-Bottleneck Perspective

**Zeyu Gan, Yong Liu***
Gaoling School of Artificial Intelligence
Renmin University of China
Beijing, China
`{zygan,liuyonggsai}@ruc.edu.cn`

## Abstract

Synthetic data has become a pivotal resource in post-training tasks for large language models (LLMs) due to the scarcity of high-quality, specific data. While various methods have been developed to generate synthetic data, there remains a discernible gap between the practical effects of synthetic data and our theoretical comprehension. To address this challenge, we commence by presenting a detailed modeling of the prevalent synthetic data generation process. Building upon this modeling, we demonstrate that the generalization capability of the post-trained model is critically determined by the information gain derived from the generative model, as analyzed from a novel reverse-bottleneck perspective. Moreover, we introduce the concept of Generalization Gain via Mutual Information (GGMI) and elucidate the relationship between generalization gain and information gain. This analysis serves as a theoretical foundation for synthetic data generation and further highlights its connection with the generalization capability of post-trained models, offering an understanding about the design of synthetic data generation techniques and the optimization of the post-training process. We open-source our code at `https://github.com/ZyGan1999/Towards-a-Theoretical-Understanding-of-Synthetic-Data-in-LLM-Post-Training`.

## 1 Introduction

The efficacy of large language models (LLMs) is extensively influenced by both the volume and quality of the training data, as established by the widely acknowledged scaling laws (Kaplan et al., 2020). Given the inherent sparsity of data available during the post-training phases of LLMs, synthetic data plays a critical role, particularly during fine-tuning and alignment processes. Over the past decades, the LLM community has increasingly employed synthetic data to augment training in scenarios where real data is scarce. As of September 2024, there are over 1,000 datasets labeled as "synthetic" on the Hugging Face platform[1]. Several leading-edge large language models, including LLaMA (Dubey et al., 2024), Falcon (Almazrouei et al., 2023), Qwen (Bai et al., 2023), and GPT-4 (OpenAI et al., 2024), have also reported utilizing synthetic data during their post-training stages. These instances underscore the pivotal role of synthetic data in enhancing the post-training of LLMs.

Numerous methodologies for synthetic data generation have been advanced (Patel et al., 2024; Møller et al., 2023; Park et al., 2024), yet the most prevalent and efficacious approach within the community involves generating synthetic data through sampling from a proficiently trained generative model, often another LLM tailored for specific domain tasks. To delineate this process more precisely, Long et al. (2024) describe the generation of synthetic data as follows: a well-trained generative model $M$ is utilized, and synthetic data $S_{\text{gen}}$ is produced by sampling from $M$, conditioned on a set of prompts $p$, just as illustrated in the lower part of Figure 1 (a).

---

*Corresponding Author.
[1]https://huggingface.co/

Synthetic data with such a generation manner is widely recognized and has been verified to be effective in LLM post-training practice. However, several challenges persist that compromise its potential benefits. First, the quality and diversity of synthetic data can vary significantly depending on the generation method and the underlying model parameters (Koo et al., 2023). This variability can lead to inconsistent training outcomes and may not fully address the sparsity in real data. Additionally, while synthetic data offers a promising solution to enrich the limited real data, ensuring that it sufficiently mimics real-world complexities without carrying over biases or errors from the original data is still a daunting task (Villalobos et al., 2022). Addressing these challenges requires a nuanced understanding of both the generation processes and their interaction with model training dynamics.

Unfortunately, there remains a significant gap in the rigorous modeling of synthetic data, which in turn limits a deeper understanding of its inherent mechanisms (Liang et al., 2024). This lack of a comprehensive theoretical framework hinders our ability to predict the effectiveness of synthetic data across different LLM applications and constrains the optimization of generative models for more targeted data synthesis (Giles et al., 2022). Consequently, advancing our knowledge on how synthetic data interacts with LLMs during training phases is crucial for enhancing model performance and reliability, and can enable the development of tailored synthetic datasets that more effectively address specific gaps in training data, thereby enhancing the overall performance and generalization capabilities of large language models.

In this paper, we endeavor to examine the influence of synthetic data on the post-training phases of large language models (LLMs) through an analytical lens focused on data distribution and information content. Our investigation seeks to address the following theoretical questions:

- What underlies the effectiveness of synthetic data? How can we model the data generation process and connect it with the generalization capabilities of post-trained models?
- What is the reason for the effectiveness of synthetic data in LLM post-training?

In response to these inquiries, we introduce a theoretical framework designed to dissect the impacts of synthetic data on LLM post-training. The principal contributions of our study are outlined as follows:

1. We develop a modeling of synthetic data generation from a distributional perspective, providing a theoretical foundation for understanding the generation process and its implications on LLM post-training.

2. Drawing on this modeling, we propose a **reverse-bottleneck framework** that elucidates the mechanisms through which synthetic data influences LLM post-training.

3. We perform a theoretical analysis from an information-theoretic standpoint, delivering several upper bounds that quantifies the expected generalization capabilities of LLMs when trained with synthetic data.

The remainder of this paper is structured as follows. In Section 2, we provide a comprehensive review of literature pertinent to our research. In Section 3, we first delineate the symbols and foundational concepts critical to our analysis, then introduce the modeling for synthetic data generation and bridge its connection with generalization capability of post-trained models. Section 4 introduces our novel reverse-bottleneck framework, designed to assess the effects of synthetic data on post-training stages of LLMs, and to establish generalization error upper bounds. The paper concludes with Section 5, summarizing our findings and discussing potential avenues for future research.

## 2 RELATED WORK

### 2.1 GENERATIVE DATA AUGMENTATION

Generative models constitute a category of machine learning models that are specifically trained to create new data points mimicking the original data distribution. Various types of generative models have been developed, each suited to particular data types and model architectures. Notable among these are Variational Autoencoders (Kingma, 2013), Generative Adversarial Networks (Goodfellow

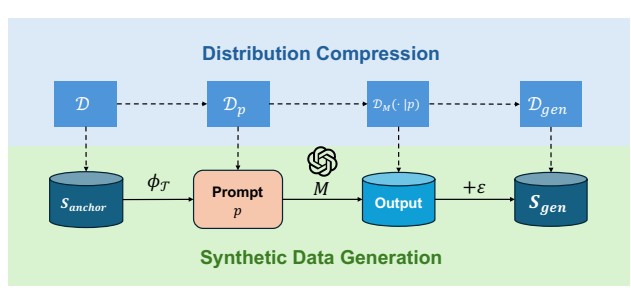
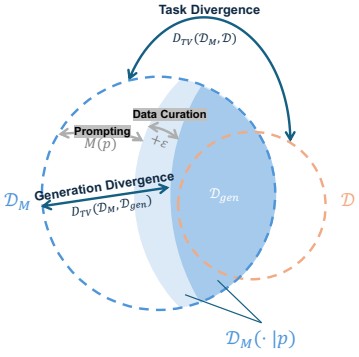

(a) Overview of synthetic data generation         (b) The relationships between distributions

Figure 1: An overview of the synthetic data generation modeling and the relationships between the distributions. **(a)** The synthetic data generation process and the corresponding distribution compression process. **(b)** The relationships between the distributions in the generation process.

et al., 2014), Normalizing Flows (Rezende & Mohamed, 2015), and, more recently, diffusion models (Rombach et al., 2022). Building on this premise, generative data augmentation has emerged as a promising approach to bolster machine learning model performance (Yamaguchi et al., 2020). This technique involves scaling up the available training dataset by generating new data points from a limited pool of labeled data using generative models. Empirical evidence suggests that generative data augmentation is particularly effective across various tasks, including knowledge graph reasoning (Maharana & Bansal, 2022), text-to-image generation (Yin et al., 2023), and relation extraction from natural language texts (Hu et al., 2023). Theoretical investigations have also been conducted to elucidate the underlying mechanisms through which generative data augmentation delivers these benefits (Zheng et al., 2023a). Collectively, these advancements highlight generative data augmentation as a highly promising avenue for improving machine learning model performance, especially in scenarios characterized by a scarcity of labeled data.

## 2.2 SYNTHETIC DATA IN LLMS

Similar to traditional generative data augmentation, synthetic data produced by LLMs is increasingly utilized to enhance post-training phases. Given the scarcity of labeled data in specific domains, synthetic data plays a crucial role in boosting the performance of LLMs across a variety of downstream tasks, including text classification (Li et al., 2023), clinical text mining (Tang et al., 2023), and code generation (Tsai et al., 2024). However, unlike classic generative data augmentation, synthetic data within LLMs is typically generated by the language models themselves and often predominates the training data in post-training stages. This predominance stems from the high-quality demands of synthetic data in LLM contexts, which necessitates alignment with human intent. Efforts to enhance the quality of synthetic data in LLMs have included integrating methodologies such as active learning (Wagner et al., 2024) and reinforcement learning (Setlur et al., 2024). Despite these advancements, the theoretical understanding of how synthetic data influences the learning process in LLMs remains limited. Key questions persist regarding the mechanisms through which synthetic data impacts LLM training and the optimal strategies for designing synthetic data to maximize LLM performance (Long et al., 2024). Addressing these questions is essential for furthering our comprehension and utilization of synthetic data in enhancing large language model efficacy.

## 2.3 INFORMATION BOTTLENECK THEORY & GENERALIZATION CAPABILITY

The information bottleneck (IB) theory, as introduced by (Tishby et al., 2000), serves as a theoretical construct designed to elucidate the learning processes within neural networks. In essence, for a given Markov chain $X \rightarrow Z \rightarrow Y$, the IB theory aims to optimize the learning process by maximizing the mutual information between $Y$ and $Z$ while minimizing the mutual information between $X$ and $Z$ (Hu et al., 2024). IB theory has been widely adopted across various deep learning fields, such as text classification (Slonim et al., 2001), sentence summarization (West et al., 2019),

and image clustering (Hu et al., 2019). Expanding upon these foundations, further research has explored generalization error upper bounds that incorporate mutual information (Russo & Zou, 2019; Xu & Raginsky, 2017). These studies have established a connection between the generalization capabilities of deep neural networks (DNNs) and IB theory (Alquier et al., 2024). More recent advancements have also highlighted the links between mutual information bounds and the PAC-Bayes framework (Banerjee & Montúfar, 2021). This type of bound suggests that the generalization error is intrinsically limited by the relevance between the training data and the learned model parameters.

## 3 PRELIMINARIES

### 3.1 NOTATIONS & EXPERIMENTAL SETUP

Let $S_{\text{anchor}}$ represent the real data utilized for generation, and $S_{\text{gen}}$ denote the synthetically generated data. The LLM employed in the generation process is designated as $M$, with the input prompt labeled as $p$. The distribution of the post-training target task $\mathcal{T}$ is referred to as $\mathcal{D}$, while the output distribution of the LLM is denoted by $\mathcal{D}_M$. Additionally, the distribution corresponding to the synthetic data is represented as $\mathcal{D}_{\text{gen}}$. The generalization error associated with the under-aligned LLM $\pi$ on the synthetic data $S_{\text{gen}}$ is expressed as $\text{Err}(\pi^{S_{\text{gen}}})$, and the generalization error related to the anchor data is indicated by $\text{Err}(\pi^{S_{\text{anchor}}})$. We define $H(\cdot)$ as the entropy of a random variable, $I(\cdot, \cdot)$ as the mutual information between two random variables, $D_{\text{KL}}$ as the Kullback-Leibler divergence, and $D_{\text{TV}}$ as the total variation distance. The detailed definitions are listed in Appendix A.1.

To provide a more intuitive demonstration, we use an example in the Gaussian mixture model (GMM) setting during the explanation. In simple terms, we assume that the target of the post-training task contains $K + J$ Gaussian distribution components, and set up a corresponding ground-truth GMM (gt-GMM, $G$) to represent the target of the post-training task. After that, we randomly sample from the first $K$ components of the gt-GMM as anchor data. To simulate the generative model $M$, we added $L$ random components to the gt-GMM, which may include extra distributions, making $M$ a GMM with total $K + J + L$ components. Finally, we randomly sampled data from $M$ to obtain the simulated synthetic data. The detailed experimental setup is listed in Appendix B.

### 3.2 MODELING SYNTHETIC DATA GENERATION

Long et al. (2024) provided a brief summary for the synthetic data generation, the overall process of synthetic data generation can be modeled as $S_{\text{gen}} \leftarrow M_p(\mathcal{T}, S_{\text{anchor}})$, where $S_{\text{gen}}$ is the generated synthetic data, $M$ is the generation model (usually a well-trained LLM), $p$ is the prompt for generation, $\mathcal{T}$ is the downstream task, and $S_{\text{anchor}}$ is the anchor data (real data). More specifically, the prompt $p$ is derived from the generation task $\mathcal{T}$ and the anchor data $S_{\text{anchor}}$, and consists of three crucial elements: $p(\mathcal{T}, S_{\text{anchor}}) \leftarrow E(e_{\text{task}}, e_{\text{condition}}, e_{\text{demo}})$, where $E$ is the prompt template, $e_{\text{task}}$ is the task element, $e_{\text{condition}}$ is the condition element, and $e_{\text{demo}}$ is the anchor data element. The conceptual framework of this modeling is straightforward. $S_{\text{gen}}$ essentially constitutes a modification of the output generated by $M$ in response to the prompt $p$, where the prompt $p$ is defined by the downstream task $\mathcal{T}$ and the anchor data $S_{\text{anchor}}$. The specifics of the generation process are thus governed by the prompt $p$ and $M$.

We enhance our understanding of synthetic data generation by reevaluating the distributional relationships among the anchor data $S_{\text{anchor}}$, the prompt $p$, and the synthetic data $S_{\text{gen}}$ produced. We postulate that the anchor data $S_{\text{anchor}}$ is sampled from distribution $\mathcal{D}$ associated with the downstream task, and the generation process is influenced by both the prompt $p$ and the generative model $M$. Consequently, $S_{\text{gen}}$ represents a modification of $M$'s output in response to the prompt $p$: $S_{\text{gen}} = M(p) + \epsilon$, where $\epsilon$ is a noise term for the measurement of revision, such as the data curation process.

The prompt $p$ is intricately linked to the downstream task $\mathcal{T}$ and the anchor data $S_{\text{anchor}}$. We postulate that $S_{\text{anchor}}$ forms the core of the prompt $p$, upon which a task-specific transformation function $\phi_{\mathcal{T}}$ is applied. Consequently, the prompt $p$ can be mathematically modeled as $p = \phi_{\mathcal{T}}(S_{\text{anchor}})$, where $\phi_{\mathcal{T}}$ is a function that maps the anchor data to the prompt, consists of all the task-relevant transformation, like the template and other customized settings for more faithful and diverse generation.

For simplicity, we note that $S_{\text{anchor}} \sim \mathcal{D}$, $p \sim \mathcal{D}_p$, $M(p) \sim \mathcal{D}_M(\cdot|p)$, and $S_{\text{gen}} \sim \mathcal{D}_{\text{gen}}$, and comprehensive details about the relationships between the distributions are listed in Appendix C. The overall synthetic data generation process in our modeling is depicted in Figure 1 (a). This illustration enhances our understanding of the connection between the generation process and distributions.

The lower part of Figure 1 (a) details the specific stages of data generation. Initially, the anchor data $S_{\text{anchor}}$ undergoes a transformation via the function $\phi_{\mathcal{T}}$ to constitute the prompt $p$, which in turn is used by the LLM $M$ to generate the synthetic data $S_{\text{gen}}$, incorporating noise $\epsilon$. The upper portion of Figure 1 (a) delineates the corresponding process of distribution shift. $S_{\text{anchor}}$ is derived from distribution $\mathcal{D}$, and the prompt $p$ emerges from distribution $\mathcal{D}_p$ conditioned on $\phi_{\mathcal{T}}$. The LLM $M$ produces the output $M(p)$ from the conditional distribution $\mathcal{D}_M(\cdot|p)$, and the final synthetic data $S_{\text{gen}}$ is sampled from $\mathcal{D}_{\text{gen}}$, representing a convolution of $\mathcal{D}_M$ and $\mathcal{D}_\epsilon$ also conditioned on $p$.

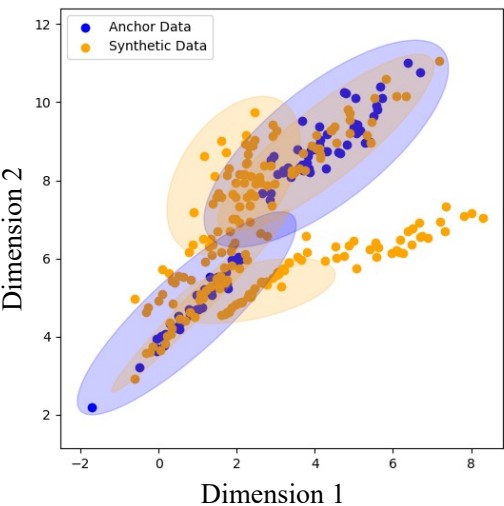

Figure 2: The simulation of the distribution relationships with GMMs. "$\bullet$" represents the anchor data sampled from distributions colored blue, and "$\bullet$" represents the synthetic data sampled from distributions colored orange.

Given that $\mathcal{D}_p$ relates solely to $\phi_{\mathcal{T}}$ and $\mathcal{D}$ (or $S_{\text{anchor}}$), and $\mathcal{D}_{\text{gen}}$ related only to $M$ and $p$, the transition from $S_{\text{anchor}}$ to $p$ to $S_{\text{gen}}$, (i.e. $S_{\text{anchor}} \rightarrow p \rightarrow S_{\text{gen}}$), constitutes a Markov chain. Figure 1 (b) provides a comprehensive view of the distributions and the nature of the distribution shift discussed. Specifically, $\mathcal{D}$ is denoted as the orange circle, and $\mathcal{D}_M$ is denoted as the blue circle. After $M$ being prompted on $p$, the conditioned distribution $\mathcal{D}_M(\cdot|p)$ is denoted as all blue areas, and the final $\mathcal{D}_{\text{gen}}$ is represented as the deep blue area after the compression on $\epsilon$. This illustration aids in understanding that **the generation process essentially compresses the output distribution of $M$, $\mathcal{D}_M$, towards the post-training target distribution $\mathcal{D}$, based on the conditions imposed by the prompt $p$ and noise $\epsilon$.**

To provide a clearer visualization, we simulate the distribution relationships using GMMs, the result is depicted in Figure 2. The distributions of $S_{\text{gen}}$ are visualized as an effort to encompass the distributions of $S_{\text{anchor}}$. However, since $S_{\text{gen}}$ is derived from the model $M$, which incorporates more complex distribution components, **the distribution of $S_{\text{gen}}$ not only attempts to mirror $S_{\text{anchor}}$ but also extends beyond, covering broader areas.**

## 3.3 BRIDGING THE GENERALIZATION CAPABILITY

Subsection 3.2 offers an exhaustive examination of the synthetic data generation process, which is pivotal for elucidating the generalization error associated with the under-aligned LLM $\pi$ when applied to synthetic data $S_{\text{gen}}$. This subsection endeavors to correlate the generalization error of $\pi$ on the synthetic data $S_{\text{gen}}$ with the synthetic data generation process as previously delineated.

Given the focus on the alignment task performance, and considering that $\pi$ is a pre-trained LLM, then is subsequently trained on synthetic data sampled from $\mathcal{D}_{\text{gen}}$, the generalization error of post-trained LLM $\pi^{S_{\text{gen}}}$ is delineated as $\text{Err}(\pi^{S_{\text{gen}}}) = \left| R_{\mathcal{D}}(\pi^{S_{\text{gen}}}) - \widehat{R}_{S_{\text{gen}}}(\pi^{S_{\text{gen}}}) \right|$, where $\mathcal{D}$ is the real distribution of the post-training task. $R_{\mathcal{D}}(\pi^{S_{\text{gen}}}) = \mathbb{E}_{z \sim \mathcal{D}} \left[ \ell(\pi^{S_{\text{gen}}}, z) \right]$ denotes the true error of $\pi^{S_{\text{gen}}}$ on the distribution $\mathcal{D}$, and $\widehat{R}_{S_{\text{gen}}}(\pi^{S_{\text{gen}}}) = \frac{1}{n} \Sigma_{z \in S_{\text{gen}}} \left[ \ell(\pi^{S_{\text{gen}}}, z) \right]$ denotes the empirical error of $\pi^{S_{\text{gen}}}$ on the synthetic data. Similar like Zheng et al. (2023b), and by the definition of the synthetic data generation process, we can simplify the above upper bound as the following lemma:

**Lemma 3.1.** *Assume that $\pi$ is with a loss function $\ell$ bounded by $C$, given an i.i.d. synthetic dataset $S_{gen}$ generated as the above defined, then the following synthetic data training generalization error*

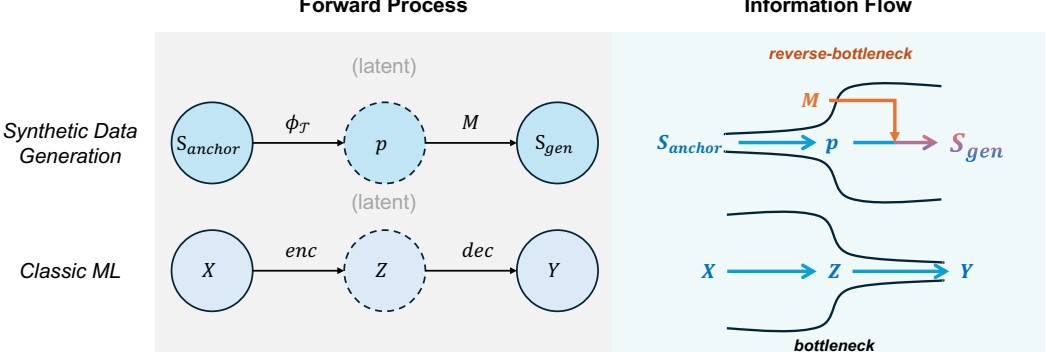

Figure 3: Illustration about the reverse bottleneck effect and comparison with classic ML process. **Left:** the similarity between the forward process of synthetic data generation and classic ML. **Right:** the difference between the information flow of the two process, where synthetic data generation gains information from $M$, constituting a reverse-bottleneck.

*upper bound holds:*

$$\text{Err}(\pi^{S_{gen}}) \leq C \underbrace{\left(D_{TV}(\mathcal{D}, \mathcal{D}_M) + D_{TV}(\mathcal{D}_M, \mathcal{D}_{gen})\right)}_{\text{Distributions' Divergence}} + \underbrace{\left|R_{\mathcal{D}_{gen}}(\pi^{S_{gen}}) - \widehat{R}_{S_{gen}}(\pi^{S_{gen}})\right|}_{\text{Generalization Error w.r.t. synthetic data}}. \quad (1)$$

The proof is referred to the Appendix D. The divergences can be defined as the task divergence ($D_{\text{TV}}(\mathcal{D}, \mathcal{D}_M)$) and the generation divergence ($D_{\text{TV}}(\mathcal{D}_M, \mathcal{D}_{gen})$), which is denoted in Figure 1 (b). The task divergence is determined by the ability of the LLM $M$ and the relevance with the task $\mathcal{T}$. The generation divergence is determined by the generation process including the prompt engineering and the data curation. In the training practice, the two divergences are controlled by either the strong ability of $M$ or the strict prompt engineering, this partially explains why synthetic data is effective.

## 4 MAIN RESULT

In this section, we delves deeper into the implications of the synthetic data generation process on the generalization capabilities.

### 4.1 INFORMATION GAIN & REVERSE-BOTTLENECK

To enhance our understanding of the synthetic data generation process, we delineate a suite of concepts pertaining to the information-flow within this process. Initially, we introduce the notion of synthetic factors, which represent the fundamental elements that influence the formation of $S_{\text{gen}}$.

**Definition 4.1.** *(Synthetic factors.) Assume that the synthetic data $S_{gen} = M(p) + \epsilon$ is derived from two factors, i.e. $M(p) = h(e_p) + g(e_M)$. The $e_p$ represents the factor w.r.t. prompt $p$ and the $e_M$ represents the factor w.r.t. applied LLM $M$.*

With the synthetic factors established, we posit that the synthetic data $S_{\text{gen}}$ is primarily governed by two distinct factors: $e_p$ and $e_M$, which are actually assumed random variables related to the prompt $p$ and the LLM $M$ respectively. Following this framework, we proceed to introduce the concept of information gain within the context of the synthetic data generation process.

**Definition 4.2.** *(Information gain.) The information gain in the synthetic data generation process is defined as:*

$$\Delta I = H(M(p)) - I\left(h(e_p), M(p)\right). \quad (2)$$

The information gain, denoted as $\Delta I$, serves as a metric for assessing the enhancement of information in the synthetic data generation process. It quantifies the incremental information content from the prompt $p$ to the synthetic data $S_{\text{gen}}$, specifically, the information introduced by the LLM $M$.

In alignment with the classical information bottleneck theory, we also introduce the concept of a compression bottleneck, which is defined in the context of synthetic factors.

**Definition 4.3.** *(Compression bottleneck.) We consider the compression bottleneck of the synthetic data towards the post-trained model parameter $W$ as:*

$$B_{syn} = I(e_M, W) + I(e_p, W). \tag{3}$$

Having delineated the concepts of information gain and compression bottleneck, we now advance our discussion to clarify the information flow within the synthetic data generation process, introducing the notion of a reverse-bottleneck effect. This framework acknowledges that the distribution $\mathcal{D}_p$ is directly influenced by $\phi_{\mathcal{T}}$ and $\mathcal{D}_{\text{anchor}}$ (or $S_{\text{anchor}}$), while $\mathcal{D}_{\text{gen}}$ pertains solely to $M$ and $p$. Consequently, the sequence $S_{\text{anchor}} \to p \to e_p \to W$ constitutes a Markov chain. Similarly, the process $M(p) \to e_M \to W$ also forms a Markov chain.

The former Markov chain, as depicted in the left part of Figure 3, parallels a classical machine learning (ML) process, in which the input $X$ is transformed into a latent representation $Z$ via an encoder, and then $Z$ is further decoded into the output $Y$ through a decoder. Similarly, in the synthetic data generation process, the input $S_{\text{anchor}}$ is converted to $p$ (which is often assumed as latent in practical applications) via $\phi_{\mathcal{T}}$, and subsequently $p$ is transformed into $S_{\text{gen}}$ by $M$. However, the presence of the latter Markov chain introduces a crucial distinction between the two processes from an information flow perspective due to the prior knowledge embedded by $M$. As illustrated in the right part of Figure 3, unlike classic ML process, **the synthetic data generation process leverages $M$ to facilitate information gains, thereby enriching the informational content of $S_{\text{gen}}$.**

This perspective emphasizes the distinctive dynamics and augmented capabilities of the synthetic data generation process in terms of capturing and utilizing information. Subsequently, we aim to analyze the relationship between the information gain and the generalization error of the model after training on the synthetic data.

## 4.2 INFORMATION-FLOW GENERALIZATION ERROR UPPER BOUND

In this subsection, we endeavor to derive the upper bounds of the generalization error from an information-flow perspective, employing the concepts previously defined. We initiate our analysis with a classical information upper bound applicable to deep neural networks, as elaborated in Lemma 4.4 (Zhang et al., 2018).

**Lemma 4.4.** *For a deep neural network with $L$ hidden layers, input $S$, and parameters $W$. The loss function is $\sigma$-sub-Gaussian with respect to $(W, Z)$ given any $w$, if all $L$ hidden layers are contraction layers, the expected generalization error can be bounded as follows,*

$$\mathbb{E}\left[R(W) - R_S(W)\right] \leq \exp\left(-\frac{L}{2}\log\frac{1}{\eta}\right)\sqrt{\frac{2\sigma^2}{n}I(S, W)}. \tag{4}$$

Lemma 4.4 establishes a connection between the expected generalization error and the mutual information between training data $S$ and learned model parameters $W$. Despite network depth $L$ and instance volume $n$, the principal constraint is imposed by the mutual information term.

Accordingly, in scenarios where post-training is with synthetic data, the generalization error is inherently constrained by the mutual information between the synthetic data $S_{\text{gen}}$ and LLM parameters after training, denoted as $I(S_{\text{gen}}, W)$. Characterizing this term presents a significant challenge due to the difficulty in measuring mutual information accurately. To address this, we introduce an analytical upper bound for $I(S_{\text{gen}}, W)$ in Lemma 4.5 to facilitate a more comprehensive understanding of the dynamics influencing model performance in post-training.

**Lemma 4.5.** *(Information-flow upper bound.) Given a synthetic dataset $S_{gen}$ defined above, and model parameters $W$ learned from $S_{gen}$, the mutual information term $I(S_{gen}, W)$ can be bounded by the following inequality:*

$$I(S_{gen}, W) \leq -\Delta I + B_{syn} + H(e_M) + \delta_{\epsilon,p}, \tag{5}$$

where $\delta_{\epsilon,p}$ indicates the efficiency during the data curation and model prompting process, which is detailed in the proof in Appendix E. Together with Lemma 4.4, we can further derive an upper bound for a training procedure with relation to the synthetic data defined above in Lemma 4.6.

**Lemma 4.6.** *(Generalization error upper bound w.r.t. synthetic data.) For a deep neural network $\pi$ with $L$ hidden layers, the parameters $W$ are optimized from synthetic data $S_{gen}$ described above. The loss function is $\sigma$-sub-Gaussian with respect to $(W, Z)$ given any $w$, if all $L$ hidden layers are contraction layers, the expected generalization error can be bounded as follows:*

$$\mathbb{E}\left|R_{\mathcal{D}_{gen}}(\pi^{S_{gen}}) - \widehat{R}_{S_{gen}}(\pi^{S_{gen}})\right| \le \exp\left(-\frac{L}{2}\log\frac{1}{\eta}\right)\sqrt{\frac{2\sigma^2\left[-\Delta I + B_{syn} + H(e_M) + \delta_{\epsilon,p}\right]}{n}}. \quad (6)$$

Lemma 4.6 delineates a quantifiable upper bound for the expected generalization error in relation to synthetic data. Beyond basic configuration parameters such as network depth $L$ and data size $n$, this upper bound is determined by four key factors outlined in the corresponding remarks.

**Remark 1.** $\Delta I$ quantifies the information gain during the data generation process. This bound demonstrates that an increase in information extracted from the model $M$ enhances the quality of the generated data.

**Remark 2.** $B_{syn}$ denotes the compression bottleneck, which is defined as the mutual information between synthetic factors and the model parameters $W$. A more pronounced compression of this term leads to improved generalization performance.

**Remark 3.** $H(e_M)$ represents the entropy associated with the synthetic factor relative to the model $M$. Intuitively, reducing this entropy by choosing a model $M$ more aligned with the specific tasks can substantially enhance downstream generalization.

**Remark 4.** $\delta_{\epsilon,p}$ concerns the efficiency during the data curation and model prompting process, highlighting the impact of noise and other data degradation factors on the overall data utility.

These factors collectively influence the generalization performance, indicating that a better generalization ability can be achieved by enhancing the information gain, reducing the compression bottleneck, minimizing the entropy, and balancing the efficiency. Finally, by integrating the insights from Lemma 3.1, the overall upper bound of the expected generalization error in the LLM post-training with synthetic data can be derived as a comprehensive boundary in Theorem 4.7.

**Theorem 4.7.** *(Synthetic data post-training upper bound.) For the same condition as lemma 4.6 and a synthetic data generation process described above, the generalization error of the model $\pi$ post-trained on the synthetic data can be bounded as:*

$$\mathbb{E}(\mathrm{Err}(\pi^{S_{gen}})) \le C\underbrace{\left(D_{TV}(\mathcal{D}, \mathcal{D}_M) + D_{TV}(\mathcal{D}_M, \mathcal{D}_{gen})\right)}_{\textit{Distributions' Divergence}}$$

$$+ \underbrace{\exp\left(-\frac{L}{2}\log\frac{1}{\eta}\right)\sqrt{\frac{2\sigma^2\left[-\Delta I + B_{syn} + H(e_M) + \delta_{\epsilon,p}\right]}{n}}}_{\textit{Generalization Error w.r.t. synthetic data}}. \quad (7)$$

## 4.3 GENERALIZATION GAIN WITH SYNTHETIC DATA

Theorem 4.7 establishes a general upper bound for the generalization error of LLMs post-trained with synthetic data. In this section, our objective is to analyze the generalization gains achieved by using synthetic data compared to scenarios devoid of synthetic data.

We commence our analysis with the anchor data $S_{anchor}$. Analogous to the definition of $\mathrm{Err}(\pi^{S_{gen}})$, the generalization error of an LLM that has been post-trained on $S_{anchor}$ is defined as $\mathrm{Err}(\pi^{S_{anchor}}) = \left|R_{\mathcal{D}}(\pi^{S_{anchor}}) - \widehat{R}_{S_{anchor}}(\pi^{S_{anchor}})\right|$. It is logically sound to assume that $S_{anchor}$ is sampled from the distribution $D$. Building upon Lemma 4.4 and assume that $S_{anchor}$ comprises $m$ instances, we can derive the subsequent result in Lemma 4.8.

**Lemma 4.8.** *(Anchor data post-training upper bound.) For the same condition as lemma 4.6, the generalization error of the model $\pi$ post-trained on the anchor data can be bounded as:*

$$\mathbb{E}(\mathrm{Err}(\pi^{S_{anchor}})) \le \exp\left(-\frac{L}{2}\log\frac{1}{\eta'}\right)\sqrt{\frac{2\sigma^2}{m}I(S_{anchor}, W')}, \quad (8)$$

where $\eta^{'}$ and $W^{'}$ are the variables of the model trained with $S_{\text{anchor}}$, noted different from that of model trained with $S_{\text{gen}}$. $\eta^{'}$ is a constant depending on the information loss and $W^{'}$ is the model parameters. Given that $m << n$ typically applies in real-world scenarios, Lemma 4.8 often represents a less stringent upper bound compared to Lemma 4.4, this results in potentially poorer generalization when relying solely on $S_{\text{anchor}}$ rather than utilizing synthetic data.

But a pertinent question arises: **do other aspects of synthetic data generation, beyond the influence of data size, also contribute to improvements in generalization performance?** Our focus is on examining how various elements within the synthetic data process impact generalization during post-training. It is inappropriate, however, to directly compare other components across these two bounds due to variations in loss and training data specifics, which affect the parameters $\eta$ and $W$ differently, where $\eta$ represents a measure of information compression and is challenging to quantify accurately (Zhang et al., 2018). Thus, our analysis primarily centers on the mutual information terms $I(S_{\text{anchor}}, W^{'})$ and $I(S_{\text{gen}}, W)$. To systematically evaluate the generalization capabilities conferred by synthetic data in relation to these mutual information metrics, we introduce a definition for generalization gain measurement as Definition 4.9.

**Definition 4.9.** *(Generalization Gain via Mutual Information, GGMI.) GGMI is defined as the difference between the mutual information terms in the two generalization upper bounds:*

$$\text{GGMI} = I(S_{anchor}, W^{'}) - I(S_{gen}, W). \tag{9}$$

A larger upper bound for the GGMI signifies greater potential generalization benefits when utilizing synthetic data. To elucidate the impact of synthetic data on model generalization, we isolate the influence of $W^{'}$ and establish that the GGMI can be effectively bounded.

**Theorem 4.10.** *(Upper bound of GGMI.) Given the synthetic data generation above, $W^{'}$ is parameterized by training with $S_{anchor}$, and $W$ is parameterized by training with $S_{gen}$, the GGMI can be bounded as follows:*

$$\text{GGMI} \leq \Delta I - (\alpha + 1)H(S_{anchor}|W) + 2\Delta H + H(S_{gen}|W) + \epsilon_{W,p}, \tag{10}$$

*where $\Delta H = H(S_{anchor}) - H(S_{gen})$, $\epsilon_{W,p} = H(S_{anchor}|W) - H(S_{anchor}|M(p))$, it is assumed that $H(S_{anchor}|W^{'}) = \alpha H(S_{anchor}|W)$, $\alpha \geq 0$.*

The proof is referred to Appendix F. Consequently, we proceed to conduct a thorough analysis of each component specified in Theorem 4.10.

**Remark 1.** $\Delta I$ represents the information gain derived from the model $M$. An increase in this information gain typically leads to improved generalization capability for $\pi^{S_{\text{gen}}}$ compared to $\pi^{S_{\text{anchor}}}$, as the model leverages additional insights to enhance performance.

**Remark 2.** $H(S_{\text{anchor}}|W)$ indicates the conditional entropy between the anchor data $S_{\text{anchor}}$ and the model parameters $W$. For a larger upper bound of GGMI, it is encouraged to decrease this value by strengthen the relevance between model parameters $W$ and anchor data $S_{\text{anchor}}$.

**Remark 3.** $\Delta H$ denotes the entropy decrease when generating synthetic data $S_{\text{gen}}$ from anchor data $S_{\text{anchor}}$. It implies that more uncertainty is eliminated during synthetic data generation leads to more generalization ability.

**Remark 4.** $H(S_{\text{gen}}|W)$ reflects the conditional entropy between the synthetic data $S_{\text{gen}}$ and the model parameters $W$. Weakening the relevance between these two entities is encouraged to ensure that the model learns the general pattern of synthetic data thus leading to better generalization.

**Remark 5.** $\epsilon_{W,p}$ denotes the effect of information compression by the training algorithm. A more pronounced compression effect typically results in a higher value, suggesting that efficient data representation contributes positively to model efficacy.

As emphasized in (Long et al., 2024), the generation of synthetic data typically focuses on two primary objectives: faithfulness and diversity. These objectives are associated with $\Delta H$ and $\Delta I$, respectively. Specifically, $\Delta H$, which quantifies the entropy decrease during synthetic data generation, as presented in Theorem 4.10, encourages the model to eliminate uncertainty during synthetic data generation, thereby enhancing the faithfulness of the synthetic data. In addition, $\Delta I$ serves as a measurement of the additional information introduced by the generative model $M$. Given that $M$ is typically pre-trained on a more extensive dataset, $\Delta I$ in Theorem 4.10 promotes the objective of diversity by facilitating greater information gain from $M$.

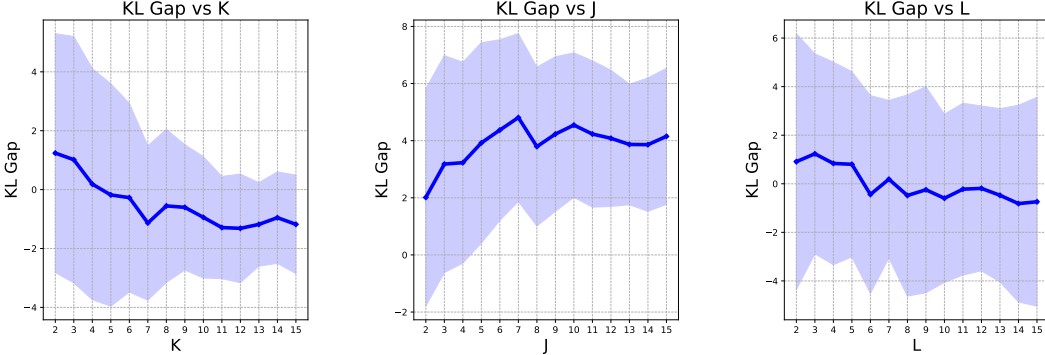

Figure 4: KL Gap with different components settings. By default, we set $K = J = L = 2$, and vary each of them from $2$ to $15$ to observe the corresponding change of KL Gap. An increase of KL Gap is observed when $J$ increases, while a decrease is observed when $K$ and $L$ increase. The shading indicates the standard deviation of 100 rounds of random settings.

## 4.4 VERIFICATION WITH GMM SIMULATION

Building upon the simulation settings, we offer a straightforward validation of the theoretical results discussed above. Specifically, we first fit a GMM $\pi$ comprising $K+J+L$ components to both $S_{\text{anchor}}$ and $S_{\text{gen}}$, yielding $\pi^{S_{\text{anchor}}}$ and $\pi^{S_{\text{gen}}}$ respectively. We then introduce a metric termed KL Gap, defined as $D_{KL}(\pi^{S_{\text{anchor}}}||G) - D_{KL}(\pi^{S_{\text{gen}}}||G)$, which represents the difference of KL-divergence between the fitted GMMs ($\pi^{S_{\text{anchor}}}$ and $\pi^{S_{\text{gen}}}$) and the ground-truth GMM $G$. A larger KL Gap corresponds to a greater GGMI, indicating enhanced generalization benefits from synthetic data.

To control the variables outlined in Theorem 4.10, we adjust the number of components in the GMM $M$ and the ground-truth GMM $G$. The result is illustrated in Figure 4. Generally, increasing $J$ facilitates the scaling of $\Delta I$, resulting in a larger upper bound for GGMI. In contrast, larger $K$ amplifies the influence of anchor data within the post-training target distribution, thereby increasing the $H(S_{\text{anchor}}|W)$ term and tightening the upper bound of GGMI. Additionally, while an increase in $L$ enhances $H(S_{\text{gen}}|W)$, it concurrently leads to a reduction in $\Delta H$. As a result, we observe a trade-off manifested as a decrease in the KL Gap in our simulation outcomes.

## 5 CONCLUSION

In this paper, we have conducted a detailed analysis of synthetic data utilization in post-training large language models (LLMs). We present a comprehensive modeling of the current synthetic data generation process, focusing on its distributional aspects, which further connects the generalization capabilities of post-trained models. We introduce a novel reverse-bottleneck framework, allowing us to derive a measurable upper bound on generalization errors. Our analysis reveals that the pivotal constraint on generalization ability is influenced by the information gain from the generative model $M$. Additionally, we present the Generalization Gain via Mutual Information (GGMI), showing that larger information gains enhance the generalization capability of post-trained models. We emphasize the importance of balancing faithfulness and diversity during post-training stages, providing a theoretical foundation for existing methodologies. Unfortunately, due to limitations in computational resources, we are unable to validate our findings within real-world LLM settings. Looking ahead, future research should focus on developing adaptive models that respond to the evolving characteristics of synthetic data. This includes enhancing generative models and fine-tuning parameters for specific learning scenarios, as well as exploring various generative models to better replicate real-world data complexities while improving model performance.

ACKNOWLEDGMENTS

This research was supported by National Natural Science Foundation of China (No.62476277), National Key Research and Development Program of China (No.2024YFE0203200), CCF-ALIMAMA TECH Kangaroo Fund (No.CCF-ALIMAMA OF 2024008), and Huawei-Renmin University joint program on Information Retrieval. We also acknowledge the support provided by the fund for building worldclass universities (disciplines) of Renmin University of China and by the funds from Beijing Key Laboratory of Big Data Management and Analysis Methods, Gaoling School of Artificial Intelligence, Renmin University of China, from Engineering Research Center of Next-Generation Intelligent Search and Recommendation, Ministry of Education, from Intelligent Social Governance Interdisciplinary Platform, Major Innovation  Planning Interdisciplinary Platform for the "Double-First Class" Initiative, Renmin University of China, from Public Policy and Decision-making Research Lab of Renmin University of China, and from Public Computing Cloud, Renmin University of China.

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

# A  DEFINITION AND INTRODUCTION ABOUT INFOMATION BOTTLENECK THEORY

## A.1  DEFINITION OF NOTATIONS

We summarize the notations used in subsection 3.1 and provide their specific definitions.

First, we define the notations about the concept related to information entropy.

**Definition A.1.** *(Entropy of a random variable.) The entropy of a random variable $X$ is defined as:*

$$H(X) = -\sum_x p(x) \log p(x),$$

*For continual random variable, the entropy is defined as:*

$$H(X) = -\int p(x) \log p(x) dx.$$

The entropy is a measurement of the uncertainty of the random variable, and the larger the entropy, the more uncertain the random variable is. It can also be considered as the average information content of the random variable.

**Definition A.2.** *(Conditional entropy of a random variable.) The conditional entropy of a random variable $X$ given another random variable $Y$ is defined as:*

$$H(X|Y) = -\sum_{x,y} p(x, y) \log p(x|y).$$

*For continual random variable, the conditional entropy is defined as:*

$$H(X|Y) = -\int p(x, y) \log p(x|y) dx dy.$$

The conditional entropy is a measurement of the uncertainty of the random variable $X$ given the information of the random variable $Y$. It can also be considered as the average information content of the random variable $X$ with $Y$ given.

Building upon the definitions above, we can further define the concepts we used in the main text with relation to information theory, including relative entropy, total variation distance, and mutual information.

**Definition A.3.** *(Relative entropy or Kullback-Leibler divergence.) The relative entropy or Kullback-Leibler divergence between two probability distributions $p$ and $q$ is defined as:*

$$D_{KL}(p\|q) = \sum_x p(x) \log \frac{p(x)}{q(x)}.$$

The relative entropy serves as a measurement of the difference between two probability distributions.

**Definition A.4.** *(Total variation distance.) The total variation distance between two probability distributions $p$ and $q$ on a finite or countable set $E$ is defined as:*

$$\begin{aligned} D_{TV}(p, q) &= \sup_{A \in E} |p(A) - q(A)| \\ &= \frac{1}{2} \sum_{x \in E} |p(x) - q(x)|. \end{aligned}$$

The total variation distance is also a measurement of the difference between two probability distributions.

**Definition A.5.** *(Mutual information.) The mutual information between two random variables $X$ and $Y$ is defined as:*

$$I(X, Y) = H(X) - H(X|Y).$$

The mutual information is a measurement of the amount of information that one random variable contains about another random variable. The larger the mutual information, the more information the two random variables share.

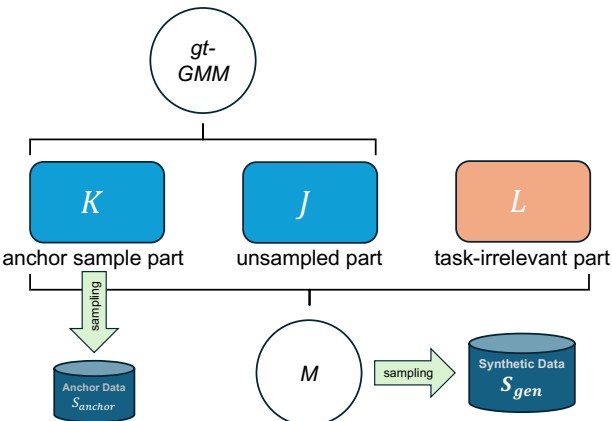

Figure 5: Illustration of the setup of the GMMs for simulation.

## A.2 THE INFORMATION BOTTLENECK THEORY

The information bottleneck (IB) theory is a theoretical construct designed to elucidate the learning processes within neural networks. In essence, for a given Markov chain $X \rightarrow Z \rightarrow Y$, the IB theory aims to optimize the learning process by maximizing the mutual information between $Y$ and $Z$ while minimizing the mutual information between $X$ and $Z$. The optimization objective in IB theory is generally expressed as:

$$\mathcal{L}\left[p\left(z|x\right)\right] = I(Z,X) - \beta I(Z,Y). \tag{11}$$

Originally developed within the context of information compression, IB theory has been widely adopted across various deep learning fields, and further research has explored generalization error upper bounds that incorporate mutual information (Russo & Zou, 2019; Xu & Raginsky, 2017). These studies have established a connection between the generalization capabilities of deep neural networks (DNNs) and IB theory (Alquier et al., 2024). A representative formulation of a generalization error upper bound from the perspective of mutual information is as follows:

$$\text{genErr} \leq \sqrt{\frac{2\sigma^2}{n} I(S,W)}, \tag{12}$$

where $S$ and $W$ are training data and model parameters respectively, with the assumption that the loss is $\sigma$-subgaussian. This type of bound suggests that the generalization error is intrinsically limited by the relevance between the training data and the learned model parameters.

## B DETAILS OF EXPERIMENTAL SETTINGS

We utilize Gaussian Mixture Models (GMMs) to simulate the data generation process, as illustrated in Figure 5. Overall, we use a gt-GMM to simulate the ground-truth, or the post-training target distribution, and a GMM $M$ to simulate the generative model applied in the data generation process.

Their are three parts of components in the GMMs: the anchor sample part with $K$ components, the unsampled part with $J$ components, and task-irrelevant part with $L$ components. It is assumed that the post-training target distribution is a combination of the anchor sample part and the unsampled part, thus the gt-GMM contains $K+J$ components from the anchor sample part and the unsampled part, which is denoted as blue in Figure 5. However, the anchor data is only sampled from the anchor sample part. This is a reasonable assumption for the real-world scenario, since the anchor data is sparse and hard to cover the whole post-training task distribution.

Additionally, the generative model $M$ is assumed to be a GMM with $K+J+L$ components. Except for the post-training target distribution, $M$ also contains a task-irrelevant part, which is denoted as orange in Figure 5. This is due to the fact that the generative model is always pre-trained on a larger scale of data, and may not be perfectly aligned with the post-training target distribution, and may introduce task-irrelevant components in the synthetic data generation process.

Building upon the settings above, we sample from the anchor sample part components of the gt-GMM to generate the anchor data $S_{\text{anchor}}$, and sample from the generative model $M$ to generate the synthetic data $S_{\text{gen}}$. In the experiment, we set the dimension of the data to be $d = 2$, and $K = J = L = 2$ by default, to facilitate the visualization and analysis of the data generation process.

For simulation in the main text, we set the number of initial anhcor data $N = 50$ for each anchor sample part component, and resample the 1000 data points for both GMM fitted on $S_{\text{anchor}}$ and $S_{\text{gen}}$. For the simulation to evaluate the KL Gap, the results are averaged over the 100 rounds, where for each round, we also resample the final data points for 100 rounds.

## C DETAILS OF SYNTHETIC DATA GENERATION MODELING

In this section, we elaborate on the modeling aspects of synthetic data generation, particularly focusing on the distributions of the prompt $p$ and synthetic data $S_{\text{gen}}$, which are central to the process of generating synthetic data for training large language models (LLMs).

**Distribution of $p$:** The prompt $p$ is is derived from the transformation function $\phi_{\mathcal{T}}$, applied to the anchor data $S_{\text{anchor}}$. This function is assumed to be reversible, allowing us to explore its properties in the context of data generation:

$$p = \phi_{\mathcal{T}}(S_{\text{anchor}}),$$

where $\phi_{\mathcal{T}}$ integrates various task-specific and conditional elements, defined as $e_{\text{task}}$ and $e_{\text{condition}}$. Assuming that $\phi_{\mathcal{T}}$ is reversible, we can derive the distribution of $p$ through the probability density function (PDF) of $\mathcal{D}_{\text{anchor}}$ (denoted as $f_{\mathcal{D}_{\text{anchor}}}$), the distribution of $p$ can be modeled as follows:

$$p \sim \mathcal{D}_p(\phi_{\mathcal{T}}) = \mathcal{D}_{\phi_{\mathcal{T}}^{-1}},$$

where the PDF of $\mathcal{D}_{\phi_{\mathcal{T}}^{-1}}$ is expressed as:

$$f_{\phi_{\mathcal{T}}^{-1}}(x) = f_{\mathcal{D}_{\text{anchor}}}(\phi_{\mathcal{T}}^{-1}(x)) \left| \det\left( \frac{\partial \phi_{\mathcal{T}}^{-1}}{\partial \mathbf{x}} \right) \right|,$$

which indicates how changes in $\mathcal{D}_{\text{anchor}}$ influence the distribution of $p$ through the transformation function, taking into account the Jacobian determinant of the inverse transformation.

**Distribution of $S_{\text{gen}}$:** The synthetic data $S_{\text{gen}}$ is the output of the large language model $M$ when prompted with $p$, typically augmented with noise $\epsilon$ to introduce variability and improve robustness. Assuming that the output of $M$ follows a specific distribution $\mathcal{D}_M$, based on the conditioning on $p$, we represent the distribution of $M(p)$ as:

$$M(p) \sim \mathcal{D}_M(\cdot \mid p),$$

The distribution of $S_{\text{gen}}$ then combines the model's output with noise, which is mathematically characterized by the convolution of $\mathcal{D}_M(\cdot|p)$ and $\mathcal{D}_\epsilon$:

$$S_{\text{gen}} \sim \mathcal{D}_{\text{gen}}(M, p) = \mathcal{D}_M(\cdot|p) * \mathcal{D}_\epsilon,$$

where $*$ is the convolution operator, integrating the noise distribution $\mathcal{D}_\epsilon$ into the output distribution of the model. This convolution reflects how noise impacts the precision and variability of the generated synthetic data, thus affecting the overall utility and effectiveness of the synthetic data in model training.

Through these detailed formulations, we aim to provide a more granular understanding of how synthetic data is modeled and generated, facilitating better integration and utilization in LLM training processes. This deeper insight into the synthetic data generation mechanics enables more targeted and effective training strategies, optimizing the performance of large language models in diverse applications.

## D PROOF OF LEMMA 3.1

*Proof.* Similar like Zheng et al. (2023b), we can further decompose the generalization error into the following three components:

$$\begin{aligned}
\text{Err}(\pi^{S_{\text{gen}}}) \leq{} & \left| R_{\mathcal{D}}(\pi^{S_{\text{gen}}}) - R_{\mathcal{D}_M}(\pi^{S_{\text{gen}}}) \right| + \left| R_{\mathcal{D}_M}(\pi^{S_{\text{gen}}}) - R_{\mathcal{D}_{\text{gen}}}(\pi^{S_{\text{gen}}}) \right| \\
& + \left| R_{\mathcal{D}_{\text{gen}}}(\pi^{S_{\text{gen}}}) - \widehat{R}_{S_{\text{gen}}}(\pi^{S_{\text{gen}}}) \right|.
\end{aligned} \tag{13}$$

For the first item in lemma, we have:

$$\left| R_{\mathcal{D}}(\pi^{S_{\text{gen}}}) - R_{\mathcal{D}_M}(\pi^{S_{\text{gen}}}) \right| = \left| \int_{\mathbf{z}} \ell(\pi^{S_{\text{gen}}}, \mathbf{z}) \left( \mathbb{P}_{\mathcal{D}}(\mathbf{z}) - \mathbb{P}_{\mathcal{D}_M}(\mathbf{z}) \right) d\mathbf{z} \right|$$

$$\leq \int_{\mathbf{z}} \left| \ell(\pi^{S_{\text{gen}}}, \mathbf{z}) \left( \mathbb{P}_{\mathcal{D}}(\mathbf{z}) - \mathbb{P}_{\mathcal{D}_M}(\mathbf{z}) \right) \right| d\mathbf{z} \tag{14}$$

$$\leq C \int_{\mathbf{z}} \left| \mathbb{P}_{\mathcal{D}}(\mathbf{z}) - \mathbb{P}_{\mathcal{D}_M}(\mathbf{z}) \right|$$

$$\lesssim C D_{\text{TV}}(\mathcal{D}, \mathcal{D}_M).$$

Similarly, for the second item in lemma, we have:

$$\left| R_{\mathcal{D}_M}(\pi^{S_{\text{gen}}}) - R_{\mathcal{D}_{\text{gen}}}(\pi^{S_{\text{gen}}}) \right| = \left| \int_{\mathbf{z}} \ell(\pi^{S_{\text{gen}}}, \mathbf{z}) \left( \mathbb{P}_{\mathcal{D}_M}(\mathbf{z}) - \mathbb{P}_{\mathcal{D}_{\text{gen}}}(\mathbf{z}) \right) d\mathbf{z} \right|$$

$$\leq \int_{\mathbf{z}} \left| \ell(\pi^{S_{\text{gen}}}, \mathbf{z}) \left( \mathbb{P}_{\mathcal{D}_M}(\mathbf{z}) - \mathbb{P}_{\mathcal{D}_{\text{gen}}}(\mathbf{z}) \right) \right| d\mathbf{z} \tag{15}$$

$$\leq C \int_{\mathbf{z}} \left| \mathbb{P}_{\mathcal{D}_M}(\mathbf{z}) - \mathbb{P}_{\mathcal{D}_{\text{gen}}}(\mathbf{z}) \right|$$

$$\lesssim C D_{\text{TV}}(\mathcal{D}_M, \mathcal{D}_{\text{gen}}).$$

Together with Eq. (13), Eq. (14), and Eq. (15), we have:

$$\text{Err}(\pi^{S_{\text{gen}}}) \leq \left| R_{\mathcal{D}}(\pi^{S_{\text{gen}}}) - R_{\mathcal{D}_M}(\pi^{S_{\text{gen}}}) \right| + \left| R_{\mathcal{D}_M}(\pi^{S_{\text{gen}}}) - R_{\mathcal{D}_{\text{gen}}}(\pi^{S_{\text{gen}}}) \right|$$

$$+ \left| R_{\mathcal{D}_{\text{gen}}}(\pi^{S_{\text{gen}}}) - \widehat{R}_{S_{\text{gen}}}(\pi^{S_{\text{gen}}}) \right|$$

$$\leq C D_{\text{TV}}(\mathcal{D}, \mathcal{D}_M) + C D_{\text{TV}}(\mathcal{D}_M, \mathcal{D}_{\text{gen}}) + \left| R_{\mathcal{D}_{\text{gen}}}(\pi^{S_{\text{gen}}}) - \widehat{R}_{S_{\text{gen}}}(\pi^{S_{\text{gen}}}) \right| \tag{16}$$

$$= C \left( D_{\text{TV}}(\mathcal{D}, \mathcal{D}_M) + D_{\text{TV}}(\mathcal{D}_M, \mathcal{D}_{\text{gen}}) \right) + \left| R_{\mathcal{D}_{\text{gen}}}(\pi^{S_{\text{gen}}}) - \widehat{R}_{S_{\text{gen}}}(\pi^{S_{\text{gen}}}) \right|.$$

This finishes the proof.

$\square$

## E  PROOF OF LEMMA 4.5

*Proof.* Considering the Markov chain $M(p) \rightarrow S_{\text{gen}} \rightarrow W$, according to the properties of mutual information, we have:

$$H(S_{\text{gen}}) \leq H(M(p)). \tag{17}$$

Furtherly, the following inequality can be derived:

$$I(S_{\text{gen}}, W) \leq I(M(p), W). \tag{18}$$

Building upon equation (18), we can derive the following equations:

$$I(S_{\text{gen}}, W) = I(M(p), W) - \delta_{\epsilon}$$

$$\leq I(M(p), W), \tag{19}$$

where $\delta_{\epsilon}$ is the information loss due to the noise $\epsilon$ in the data curation process.

Since $h(\cdot)$ and $g(\cdot)$ are deterministic functions which decrease the entropy of random variables, we have:

$$H(h(e_p)) \leq H(e_p), \quad H(g(e_M)) \leq H(e_M). \tag{20}$$

Accordingly, the following inequalities can be derived:

$$I(h(e_p), W) = H(h(e_p)) - H(h(e_p)|W)$$

$$\leq H(e_p) - H(e_p|W) \tag{21}$$

$$= I(e_p, W).$$

Similarly, we have:

$$\begin{aligned}
I(g(e_M), W) &= H(g(e_M)) - H(g(e_M)|W) \\
&\leq H(e_M) - H(e_M|W) \\
&= I(e_M, W).
\end{aligned} \tag{22}$$

This is because the deterministic functions $h(\cdot)$ and $g(\cdot)$ decrease the information content, and make the information a subset of the original random variables.

Then we consider the upper bound of $I(M(p), W)$ according to the result above:

$$\begin{aligned}
I(M(p), W) &= I(h(e_p) + g(e_M), W) \\
&\leq I(h(e_p), W) + I(g(e_M), W) \\
&\leq I(e_p, W) + I(e_M, W)
\end{aligned} \tag{23}$$

For further analysis, we consider the following assumption related to the efficiency of the model utilizing the prompt:

**Lemma E.1.** *(Efficiency of the model prompting.) For the model $M$ utilizing the prompt $p$, with $\lambda \geq 1$, we have:*

$$H(e_p) \leq \lambda I(e_p, M(p)). \tag{24}$$

Lemma E.1 indicates that the entropy of $e_p$ is upper bounded by the mutual information between the synthetic factor $e_p$ and the model output $M(p)$ by a factor of $\lambda$. In other words, the efficiency of the model utilizing the prompt is reflected in the value of $\lambda$, which quantifies the extent to which the model can leverage the information contained in the prompt. For example, a larger $\lambda$ indicates a smaller $I(e_p, M(p))$, which implies that the $M(p)$ contains more information from the prompt $p$.

Building upon Lemma E.1, we can further derive the deduction following equation (23):

$$\begin{aligned}
I(M(p), W) &\leq I(e_p, W) + I(e_M, W) \\
&= H(e_p) - H(e_p|W) + I(e_M, W) \\
&= H(M(p)) - H(M(p)) + H(e_p) - H(e_p|W) + I(e_M, W) \\
&\leq -H(M(p)) + I(e_p, M(p)) - I(e_p, M(p)) \\
&\quad + \lambda I(e_p, M(p)) + H(M(p)) - H(e_p|W) + I(e_M, W) \\
&\leq -\Delta I + I(e_M, W) + H(M(p)) - H(e_p|W) + (\lambda - 1)I(e_p, M(p)) \\
&\leq -\Delta I + B_{\text{syn}} + H(e_M).
\end{aligned} \tag{25}$$

**Lemma E.2.** *(Entropy gap upper bound) The difference between the entropy of $M(p)$ and $e_p$ can be upper bounded by the following inequality:*

$$H(M(p)) - H(e_p) \leq H(e_M). \tag{26}$$

The proof of Lemma E.2 is listed in equation (27):

$$\begin{aligned}
H(M(p)) - H(e_p) &= H(h(e_p) + g(e_M)) - H(e_p) \\
&\leq H(e(p)) + H(g(e_M)) - H(e_p) \\
&\leq H(e_p) + H(e_M) - H(e_p) \\
&= H(e_M).
\end{aligned} \tag{27}$$

Building upon Lemma E.2, we can further deduce the following inequality following equation (25):

$$\begin{aligned}
I(M(p), W) &\leq -\Delta I + I(e_M, W) + H(M(p)) - H(e_p|W) + (\lambda - 1)I(e_p, M(p)) \\
&\leq -\Delta I + B_{\text{syn}} - I(e_p, W) + H(M(p)) - H(e_p|W) + (\lambda - 1)I(e_p, M(p)) \\
&= -\Delta I + B_{\text{syn}} + H(M(p)) - H(e_p) + (\lambda - 1)I(e_p, M(p)) \\
&\leq -\Delta I + B_{\text{syn}} + H(e_M) + (\lambda - 1)I(e_p, M(p)).
\end{aligned} \tag{28}$$

Together with equations (19) and (28), we have:

$$\begin{aligned}
I(S_{\text{gen}}, W) &= I(M(p), W) - \delta_\epsilon \\
&\leq -\Delta I + B_{\text{syn}} + H(e_M) + (\lambda - 1)I(e_p, M(p)) - \delta_\epsilon \\
&\leq -\Delta I + B_{\text{syn}} + H(e_M) + \delta_{\epsilon, p},
\end{aligned} \tag{29}$$

where $\delta_{\epsilon,p} = (\lambda - 1)I(e_p, M(p)) - \delta_\epsilon$.

This finishes the proof. $\qquad\square$

## F    PROOF OF THEOREM 4.10

*Proof.* Considering the Markov chain $h(e_p) \to M(p) \to S_{\text{gen}}$, we have:

$$H(M(p)) \geq H(S_{\text{gen}}). \tag{30}$$

In addition, according to the properties of mutual information, we have:

$$I(S_{\text{anchor}}, M(p)) \geq I\left(h(e_p), M(p)\right). \tag{31}$$

Building upon the inequalities above, we can derive the following equations:

$$\begin{aligned}
\Delta I &= H(M(p)) - I\left(h(e_p), M(p)\right) \\
&\geq H(S_{\text{gen}}) - I(S_{\text{anchor}}, M(p)) \\
&= I(S_{\text{gen}}, W) + H(S_{\text{gen}}|W) - I(S_{\text{anchor}}, M(p)).
\end{aligned} \tag{32}$$

Based on the assumptions mentioned above, we also have:

$$\begin{aligned}
I(S_{\text{anchor}}; W') &= H(S_{\text{anchor}}) - H(S_{\text{anchor}}|W') \\
&= H(S_{\text{anchor}}) - \alpha H(S_{\text{anchor}}|W) \\
&= I(S_{\text{anchor}}, W) + (1 - \alpha)H(S_{\text{anchor}}|W).
\end{aligned} \tag{33}$$

Furthermore, based on the definitions, we have:

$$\begin{aligned}
I(S_{\text{anchor}}, M(p)) &= H(S_{\text{anchor}}) - H(S_{\text{anchor}}|M(p)) \\
&= I(S_{\text{anchor}}, W) + H(S_{\text{anchor}}|W) - H(S_{\text{anchor}}|M(p)) \\
&= I(S_{\text{anchor}}, W) + \epsilon_{W,p}.
\end{aligned} \tag{34}$$

By the definition of GGMI, and with equation (33), the following result can be deduced:

$$\begin{aligned}
\text{GGMI} =& I(S_{\text{anchor}}, W') - I(S_{\text{gen}}, W) \\
=& I(S_{\text{anchor}}, W) + (1 - \alpha)H(S_{\text{anchor}}|W) - I(S_{\text{gen}}, W) \\
=& I(S_{\text{gen}}, W) + H(S_{\text{gen}}|W) - I(S_{\text{anchor}}, M(p)) \\
& - I(S_{\text{gen}}, W) - H(S_{\text{gen}}|W) + I(S_{\text{anchor}}, M(p)) \\
& + I(S_{\text{anchor}}, W) + (1 - \alpha)H(S_{\text{anchor}}|W) - I(S_{\text{gen}}, W).
\end{aligned} \tag{35}$$

Subsequently, together with equations (32) and (34), we can further deduce that:

$$\begin{aligned}
\text{GGMI} \leq& \Delta I - 2I(S_{\text{gen}}, W) - H(S_{\text{gen}}|W) + I(S_{\text{anchor}}, W) \\
& + (1 - \alpha)H(S_{\text{anchor}}|W) + I(S_{\text{anchor}}, M(p)) \\
=& \Delta I - 2I(S_{\text{gen}}, W) - H(S_{\text{gen}}|W) \\
& + 2I(S_{\text{anchor}}, W) + (1 - \alpha)H(S_{\text{anchor}}|W) + \epsilon_{W,p} \\
=& \Delta I - 2H(S_{\text{gen}}) + H(S_{\text{gen}}|W) \\
& + 2H(S_{\text{anchor}}) - (\alpha + 1)H(S_{\text{anchor}}|W) + \epsilon_{W,p}.
\end{aligned} \tag{36}$$

Finally, together with all the deduce and definition above, we have:

$$\text{GGMI} \leq \Delta I - (\alpha + 1)H(S_{\text{anchor}}|W) + 2\Delta H + H(S_{\text{gen}}|W) + \epsilon_{W,p}, \tag{37}$$

This finishes the proof. $\qquad\square$

# G    EXPERIMENTS: EXPLORING BETTER SYNTHETIC DATA IN PRACTICE

To further investigate the process of synthetic data generation in real-world settings, we conduct experiments to evaluate the quality of synthetic data produced under different conditions and aim to identify the factors that contribute to its effectiveness in enhancing model performance.

The experiments follow the same setup as described in the main text, with the synthetic data $S_{\text{gen}}$ generated from a generative model $M$ prompted by $p$. We utilize a standard in-context learning (ICL) framework to determine $p$ using anchor data $S_{\text{anchor}}$, and we then evaluate the performance of the model trained on the synthetic data. Additionally, we estimate key components from our theoretical analysis in the main text, including information gain $\Delta I$ and the entropy of the synthetic data $H(S_{\text{gen}})$.

In the remainder of this section, we commence by introducing the experimental setup and the evaluation metrics. We then present the results of the synthetic data generation, focusing on the performance of the model trained on synthetic data to assess its quality. Furthermore, we estimate the key components from our theoretical analysis and analyze the factors that contribute to the effectiveness of synthetic data in improving model performance. Finally, we provide a brief conclusion and discuss potential principles for generating higher-quality synthetic data in practice.

## G.1    EXPERIMENTAL SETUP

We conducted experiments to evaluate the effectiveness of synthetic data generated by a generative model $M$ prompted by $p$ in enhancing model performance. Our experimental setup follows the synthetic data utilization process described in the main text, including selecting benchmark dataset, determining prompt $p$, generating synthetic data $S_{\text{gen}}$, training the model on the synthetic data, and evaluating the trained model.

### G.1.1    BENCHMARK DATASET

The benchmark dataset is utilized to sample $S_{\text{anchor}}$. Specifically, we adopt Dolly-15K (Conover et al., 2023) as our benchmark dataset, which contains 15,000 lines of text data designed for instruction-following tasks. We split the benchmark dataset into training and testing sets with a ratio of 8:2, and $S_{\text{anchor}}$ is sampled from the training set. For each data instance, we retain the keys "instruction", "context" and "response" and combine them using the following template.

### G.1.2    DETERMINING PROMPT $p$

Consistent with the methodology described in the main text, we employ a standard In-Context Learning (ICL) framework to determine the prompt $p$. Specifically, $p = E(S_{\text{anchor}})$, where $E$ is a predefined template for the prompt. We follow the settings of Alpaca (Taori et al., 2023) and modify the template to better suit the benchmark dataset used in our experiments. The modified template is as follows:

> You are asked to come up with a set of 20 diverse task instructions. These task instructions will be given to a language model and we will evaluate the language model for completing the instructions. Here are the requirements: 1. Try not to repeat the verb for each instruction to maximize diversity. 2. The language used for the instruction also should be diverse. For example, you should combine questions with imperative instrucitons. 3. The type of instructions should be diverse. The list should include diverse types of tasks like open-ended generation, classification, editing, etc. 4. The language model should be able to complete the instruction. For example, do not ask the assistant to create any visual or audio output. For another example, do not ask the assistant to wake you up at 5pm or set a reminder because it cannot perform any action. 5. The instructions should be in English. 6. The instructions should be 1 to 2 sentences long. Either an imperative sentence or a question is permitted. 7. You should generate an appropriate input to the instruction. The input field should contain a specific example provided for the instruction. It should involve realistic data and should not contain simple placeholders. The input should provide substantial content to make the instruction challenging but should ideally not exceed 100 words. 8. Not all instructions require input. For example, when a instruction asks about some general information, "what is the highest peak in the world", it is not necssary to provide a specific context. In this case, we simply put "noinput" in the input field. 9. The output should be an appropriate response to the instruction and the input. Make sure the output is less than 100 words.
>
> Your output should consist of 3 parts: instruction, context and reference response. "Instruction" is the task instruction which the language model should complete. "Context" is information related to the instruction, if don't need, you can set it as empty. "Reference response" is the correct answer to the instruction your recommend.
>
> Your output must be in the following json form like: {"instruction": [the instruction you generate], "context": [the context you generate], "reference response": [the reference response you generate]}
> Here are some examples you should emulate:
> **{anchor data}**
> List of 20 tasks:'''

We then sample $S_{\text{anchor}}$ from the benchmark dataset and populate the "{anchor data}" placeholders in the prompt template with these samples. This completes the process of determining the prompt $p$.

### G.1.3 GENERATING SYNTHETIC DATA $S_{\text{GEN}}$

After determining the prompt $p$, we generate synthetic data $S_{\text{gen}}$ by prompting the generative model $M$ with $p$. In our experiments, we primarily utilize GPT-4o (OpenAI, 2024) as the generative model $M$. Additionally, wo also employ the latest Llama 3.2 models (Meta, 2024) including Llama-3.2-1B-Instruct and Llama-3.2-3B-Instruct for comparison experiments.

### G.1.4 TRAINING ON SYNTHETIC DATA

We fine-tune a GPT-2 (Radford et al., 2019) model using both the synthetic data $S_{\text{gen}}$ generated by the generative model $M$ and the training set $T$ of the benchmark dataset. The training procedure follows the standard instruction tuning process, where we fine-tune the model on the synthetic data for a fixed 20 epochs.

### G.1.5 EVALUATING FINE-TUNED MODEL

We assess the performance of the fine-tuned model on the testing set of the benchmark dataset. Following the evaluation procedure of Zheng et al. (2023c), we evaluate the model's ability by rating the generated responses using a LLM. To better align our evaluation with our datasets, we modify the original evaluation prompt to ensure that the judge LLM compares the output with the ground-truth answer. The evaluation prompt we adopt is as follows:

| | **Base** | **Synthetic Data Fine-Tuned** | | | **Real Data Fine-Tuned** |
|---|---|---|---|---|---|
| | | 3-ins | 10-ins | 20-ins | |
| **Rating** | 0.1409 | 0.1863 | 0.1965 | 0.2015 | 0.2745 |

Table 1: Average ratings of the fine-tuned model on the testing set. The ratings were normalized using a softmax function. The synthetic data were generated by GPT-4o with varying numbers of instances in In-Context Learning (ICL) (denoted as $x$-ins). The unfine-tuned base model (Base) and the model fine-tuned on real data are marked gray.

> Please act as an impartial judge and evaluate the quality of the response provided by an AI assistant to the user question displayed below. You are provided with 4 parts of the text, [Question] is the question asked by the user, [Context] is information related to the question, [Reference Answer] is the correct answer for your reference, and Assistant's Answer which is surrounded by [The Start of Assistant's Answer] and [The End of Assistant's Answer] is the answer given by the assistant. Your evaluation should consider factors such as the helpfulness, relevance, accuracy, depth, creativity, and level of detail of the response. Begin your evaluation by providing a short explanation. Be as objective as possible. After providing your explanation, you must rate the Assistant's Answer on a scale of 1 to 10 by strictly following this format: "[[rating]]", for example: "Rating: [[5]]".
> [Question] **{instruction}**
> [Context] **{context}**
> [Reference Answer] **{reference response}**
> [The Start of Assistant's Answer] **{generated response}** [The End of Assistant's Answer]

We then populate the placeholders "{instruction}", "{context}", "{reference response}", and "{generated response}" in the evaluation prompt with the corresponding text. We adopt Llama-3.1-Nemotron-70B-Instruct-HF (Wang et al., 2024) as the judge LLM and extract the ratings from its output. The final rating is averaged over the testing set to evaluate the performance of the fine-tuned model.

## G.2 SYNTHETIC DATA QUALITY

We assess the quality of synthetic data generated by the generative model $M$ prompted by $p$ in terms of its effectiveness in enhancing model performance. Specifically, we utilize GPT-4o as $M$ to generate synthetic data with varying numbers of instances in ICL, corresponding to different sizes of $S_{\text{anchor}}$, denoted as 3-ins, 10-ins, and 20-ins, respectively. We then fine-tune a GPT-2 model on both the synthetic data and the training set of the benchmark dataset. The performance of the fine-tuned model on the testing set is used as a measure of the quality of the synthetic data. For better presentation, we apply a softmax function to normalize the ratings. The results are shown in Table 1.

The results demonstrate that the synthetic data effectively enhances the performance of the fine-tuned model, with the rating positively correlated with the number of instances in ICL. This finding indicates that appropriately increasing the number of instances in ICL can improve the quality of the synthetic data. This phenomenon may be attributed to the fact that increasing the number of instances in the ICL prompts provides the generative model with a richer and more diverse context. This enhanced context allows the model to capture a broader range of patterns present in the anchor data, thereby generating synthetic data with richer content.

## G.3 ESTIMATING THEORETICAL COMPONENTS

Building upon the results of synthetic data quality, we further estimate the key components from our theoretical analysis, including information gain $\Delta I$ and the entropy of the synthetic data $H(S_{\text{gen}})$. We aim to analyze the factors that contribute to improving the quality of synthetic data.

### G.3.1 ESTIMATING INFORMATION GAIN

Given the definition of information gain $\Delta I$ in Definition 4.2, it is difficult to directly estimate $\Delta I$ in practice. However, it is possible to estimate $I(T, S_{\text{gen}})$, the mutual information between the synthetic

| $S_{\text{gen}}$ | 3-ins | 10-ins | 20-ins |
|---|---|---|---|
| **HSIC w/ $T$** $_{(\times 10^{-3})}$ | 7.8703 | 7.8668 | 7.8502 |

Table 2: The HSIC value between the synthetic data $S_{\text{gen}}$ and training set $T$ for different numbers of instances in ICL setting.

| $S_{\text{gen}}$ | 3-ins | 10-ins | 20-ins |
|---|---|---|---|
| **Semantic Entropy** | 1.0739 | 1.0503 | 1.0005 |

Table 3: The semantic entropy of the synthetic data $S_{\text{gen}}$ in different numbers for instances in ICL setting.

data $S_{\text{gen}}$ and the training set $T$ of the benchmark dataset where $S_{\text{anchor}}$ is sampled. Since the crucial part of prompt $p$ is $S_{\text{anchor}}$, $I(T, S_{\text{gen}})$ has a negative correlation with $\Delta I$ to a certain extent.

To measure $I(T, S_{\text{gen}})$, we follow the setting of existing works (Qian et al., 2024; Ma et al., 2020) and utilize HSIC (Gretton et al., 2005) as an estimator. The result is shown in Table 2.

It is supurising that more instances doesn't increase the HSIC value, but even lead to a lower HSIC value, indicating reduced mutual information between the synthetic data and the training set. This phenomenon suggests that enlarging the sizes of $S_{\text{anchor}}$ does not significantly increase the dependency between the synthetic data and the training set, and may even enhance the diversity of the synthetic data. This may be attributed to the fact that when a LLM with a wide range of knowledge is employed as $M$, it leverages its broad understanding to generate synthetic data that is less reliant on the specific instances in $S_{\text{anchor}}$. As the number of instances in the ICL setup increases, the LLM interprets this as a richer and more varied context, thereby increasing the output diversity instead.

A smaller HSIC value indicates a lower mutual information between the synthetic data and the training set, which leads to a larger information gain $\Delta I$. With Theorem 4.7 and Theorem 4.10, this guarantees a tighter upper bound of the generalization error and higher GGMI, which contributes to the quality of synthetic data and increase the generalization capabilities.

### G.3.2 Estimating Entropy of Synthetic Data

As another important component in Theorem 4.10, $H(S_{\text{gen}})$ is crutial for determining the value of $\Delta H$. We use semantic entropy (Farquhar et al., 2024) as an estimator to measure the entropy of the dataset and estimate the value of $H(S_{\text{gen}})$. The result is shown in Table 3.

The results indicate that the semantic entropy of the synthetic data $S_{\text{gen}}$ is also negatively correlated with the number of instances in ICL. This suggests that increasing the sizes of $S_{\text{anchor}}$ when utilizing LLM as generative model $M$ can help reduce the entropy of the synthetic data. This reduction in entropy may be attributed to the richer and more varied context provided by a larger $S_{\text{anchor}}$, which enables $M$ to generate more accurate and informative synthetic data, thereby increasing the faithfulness of the synthetic data.

A smaller semantic entropy indicates a lower entropy of the synthetic data $S_{\text{gen}}$, which leads to a larger $\Delta H$. With Theorem 4.10, this benifts increasing the upper bound of GGMI, and contributes to the generalization capabilities of the model trained on the synthetic data.

### G.4 Estimating on Different Model Architectures

To further investigate the impact of different model architectures and parameters on the quality of synthetic data, we conduct experiments to evaluate the HSIC value and semantic entropy of the synthetic data $S_{\text{gen}}$ generated by different models. Due to computational resource limitations, we utilized GPT-4o, Llama-3.2-3B-Instruct, and Llama-3.2-1B-Instruct as the generative model $M$ to generate synthetic data with 3 instances in ICL setting. The results are presented in Table 4.

Note that under the prompt determined in the experimental setups, the Llama-3.2-1B-Instruct model did not adhere to the format requirements and failed to produce meaningful synthetic data. Consequently, the estimators are not available for this model. This observation underscores a fundamental

|  | GPT-4o | Llama-3.2-3B-Instruct | Llama-3.2-1B-Instruct |
|---|---|---|---|
| **HSIC** w/ $T_{(\times 10^{-3})}$ | 7.8703 | 11.4306 | / |
| **Semantic Entropy** | 1.0739 | 2.9427 | / |

Table 4: The HSIC value and semantic entropy of the synthetic data $S_{\text{gen}}$ generated using different model architectures. All the synthetic data are generated with 3 instances in ICL setting. Note that the Llama-3.2-1B-Instruct model did not adhere to the format requirements and thus failed to produce meaningful synthetic data.

premise that the generative model $M$ must possess sufficient instruction-following capabilities to generate synthetic data that can be effectively utilized to enhance model performance.

On the other hand, although Llama-3.2-3B-Instruct produced usable synthetic data, its quality was insufficient for fine-tuning GPT-2, and the HSIC value and semantic entropy were significantly higher than those of GPT-4o. This may be attributed to the smaller model size of Llama-3.2-3B-Instruct compared to GPT-4o, resulting in a diminished capacity to generate synthetic data that is both faithful to and diverse from the anchor data. For instance, we provide some examples of the synthetic data generated by Llama-3.2-3B-Instruct in the following as a case study:

> **Instructions generated by Llama-3.2-3B-Instruct:**
> "instruction": "Explain the concept of blockchain in simple terms."
> "instruction": "Explain the concept of artificial intelligence in simple terms."
> "instruction": "Explain the concept of climate change in simple terms."
> . . .
> "instruction": "Identify the type of music genre: classical or jazz: 'Moonlight Sonata' or 'Take Five'"
> "instruction": "Identify the type of literary device used in the following sentence: 'The eyes that fixed upon her were like two bright stars in the night sky.'"
> "instruction": "Identify the type of music instrument: string or woodwind: 'Violin' or 'Flute'"
> . . .
> "instruction": "Write a short story about a character who discovers a hidden world within their reflection."
> "instruction": "Write a review of the movie 'The Shawshank Redemption'."
> "instruction": "Write a poem about the beauty of nature."

The examples demonstrate that the synthetic data generated by Llama-3.2-3B-Instruct is highly homogeneous, even within a single generation cycle. Moreover, it is highly dependent on the specific instances in the anchor data, leading to a higher HSIC value. Furthermore, although the synthetic data lacks diversity in form, the semantic entropy remains high. This indicates that the generated synthetic data lacks sufficient faithfulness. Collectively, these factors contribute to the poor quality of the synthetic data produced by Llama-3.2-3B-Instruct.

### G.5 CONCLUSION

Building upon the experiments, we can derive some brief conclusions about how to guarantee the synthetic data quality and estimate the key factors in real-world LLM practice.

The quality of synthetic data is mainly reflected in two aspects: diversity and faithfulness. Diversity makes the synthetic data contain richer contents and thus increase the information gain $\Delta I$. With our theoretical analysis, this will benifit the generalization ability of the model post-trained on synthetic data. Faithfulness makes the synthetic data semantically continuous, and thus decrease the entropy of the synthetic data $S_{\text{gen}}$, which also strenghten the generalization capabilities.

In practice, the diversity and the faithfulness can be estimated by HSIC value and the semantic entropy, respectively, as demonstrated in the experimental settings of this section. It is also important to highlight that employing a generative model with stronger instruction-following capabilities and more diverse knowledge can enhance the quality of synthetic data in both aspects.

