# OpenReview forum: "Towards a Theoretical Understanding of Synthetic Data in LLM Post-Training: A Reverse-Bottleneck Perspective"
_ICLR.cc/2025/Conference — ICLR 2025 Poster_

### Official Review · Reviewer_7KR4 · 2024-10-21

**Soundness:** 3
**Presentation:** 3
**Contribution:** 4
**Rating:** 6
**Confidence:** 3

**Summary:**

This paper explores the use of synthetic data in post-training tasks for large language models. It presents a theoretical framework to analyze the effects of synthetic data on model generalization, focusing on the reverse-bottleneck effect. The authors introduce key metrics like Generalization Gain via Mutual Information (GGMI) and propose a detailed modeling of the synthetic data generation process. They also provide upper bounds on generalization error based on information theory.

**Strengths:**

1. The paper provides a detailed theoretical analysis of the introduction of synthetic data, clearly explaining the information gain that synthetic data can bring.
2. The authors present thorough and well-supported mathematical proofs to substantiate their claims, offering a solid foundation for their arguments.
3. The reverse-bottleneck theory mathematically captures the essence of synthetic data's impact, offering valuable insights and guidance for the development of synthetic data methodologies.

**Weaknesses:**

1. The paper primarily focuses on theoretical analysis with minimal experimental validation. It is recommended to include empirical experiments to support the proposed theory.
2. The use of GMM models and non-real numerical data raises questions about whether the findings can effectively reflect real-world LLM outcomes.
3. The theory mainly emphasizes the performance benefits of introducing synthetic data, but lacks analysis in other areas. For example, it could explore potential drawbacks of over-reliance on synthetic data in LLMs, as highlighted in studies like https://www.nature.com/articles/s41586-024-07566-y.

**Questions:**

1. How would synthetic data that has not been processed by an LLM impact the proposed theory? For instance, data generated through specific rules, such as template-based concatenation or converting game-playing sequences into text.
2. Does this theoretical analysis still hold for non-text, multimodal data? Would the same principles apply, or are there limitations in extending the theory to such data types?

---

> ### Author Response · Authors · 2024-11-20
> **Response Part 1**
>
> We would like to express our sincere gratitude for your valuable feedback and constructive suggestions on our paper. Thanks for your appreciation and advice!
>
> We noticed that all the reviewers suggest more experimental explorations on real-world settings, we would like to first introduce our enriched experiments, and then respond to your suggestions and questions.
>
> # Experiments on Real-World Settings
>  **The detailed presentation has been included in the Appendix G of our newly submitted PDF file.** We recommend referring to the paper for a thorough review. However, we still provide a brief summary of our experiments here (in the order of "generation", "training", and "evaluation").
>
> *Due to limitations in both training resources and time, the additional experiments may still be imperfect as a simulation of real-world LLMs. However, we have made every effort to present our analysis to the best of our ability. Training with LLMs is an extremely resource-consuming task, and we sincerely appreciate your understanding.*
>
> ## 0. Experimental Setup
> We follow the same setup as described in the main text, where the synthetic data $S_{\text{gen}}$ generated from a generative model $M$ prompted by $p$. We utilize a standard in-context learning (ICL) framework to determine $p$ using anchor data $S_{\text{anchor}}$, and we then evaluate the performance of the model trained on the synthetic data. Additionally, we estimate key components from our theoretical analysis in the main text, including information gain $\Delta I$ and the entropy of the synthetic data $H(S_{\text{gen}})$.
>
> ## 1. Generate Synthetic Data
> We apply Dolly-15K [1] as the benchmark dataset to sample $S_{\text{anchor}}$, then we modify the template from Alpaca [2] for determining the prompt $p$ for generating synthetic data. We adopt LLMs including GPT-4o [3], and latest Llama 3.2 models [4] like Llama-3.2-1B-Instruct and Llama-3.2-3B-Instruct as generative model $M$.
>
> By setting different numbers of instances of the ICL, we use GPT-4o to generate three datasets, noted as 3-ins, 10-ins and 20-ins respectively.
> GPT-4o Synthetic Dataset | 3-ins | 10-ins | 20-ins
> :------:|:---:|:---:|:---:
> **instance of ICL** | 3 | 10 | 20
>
> While for the two Llama 3.2 model, we generate one synthetic dataset respectively, in the setting of 3-ins.
>
> All the 5 synthetic datasets we generate are in the same size with benchmark datasets.
>
> ## 2. Fine-Tune with Synthetic Data
> We use the formerly GPT-4o generated synthetic datasets and the benchmark dataset to fine-tune a GPT-2 [5] model.
> The models are trained following a standard instruction-tuning framework, with fixed epochs to completely converge.
>
> ## 3. Evaluating the Quality of Synthetic Data
> We follow the framework of [6] and use Llama-3.1-Nemotron-70B-Instruct-HF [7] as the judge LLM to evaluate the fine-tuned models on the testing set of the benchmark dataset, and extract the ratings from its output. The final rating is averaged over the testing set to evaluate the performance of the fine-tuned model. After the nomalization, we use the ratings as a measure of the synthetic data quality.
> $S_{\text{gen}}$ | Base | 3-ins | 10-ins | 20-ins | real training set
> :----:|:------:|:-------:|:--------:|:--------:|:---:
> **Rating** | 0.1409 | 0.1836 | 0.1965 | 0.2015 | 0.2745
>
> The "Base" represents the unfine-tuned model.
>
> **conclusion & analysis**:
> - synthetic data quality is positively correlated with the number of instances in ICL.
> - appropriately increasing the number of instances in ICL can improve the quality of the synthetic data.
> - increasing the number of instances in the ICL prompts provides the generative model with a richer and more diverse context.
> - enhanced context allows the model to capture a broader range of patterns present in the anchor data, thereby generating synthetic data with richer content.
>
>
> ## 4. Estimate Components in Our Bound
> We primarily estimate the $\Delta I$ and $\Delta H$ in our bounds.
>
> For the information gain $\Delta I$, we follow [8] and [9] and use HSIC as an estimator of the mutual information between $S_{\text{gen}}$ and the benchmark training set. A larger HSIC indicates stronger relevance, and thus less information gain $\Delta I$. We calculate the HSIC value ($\times 10^{-3}$) of the GPT-4o generated synthetic datasets and the results are as follows:
> $S_{\text{gen}}$ | 3-ins | 10-ins | 20-ins
> :--:|:--:|:--:|:--:
> **HSIC** | 7.8703 | 7.8668 | 7.8502

---

> ### Author Response · Authors · 2024-11-20
> **Response Part 2**
>
> **conclusion & analysis**
> - more instances doesn't increase the HSIC value, but even lead to a lower HSIC value, indicating reduced mutual information between the synthetic data and the training set.
> - enlarging the sizes of $S_\text{anchor}$ does not significantly increase the dependency between the synthetic data and the training set, and may even enhance the diversity of the synthetic data.
> - when a LLM with a wide range of knowledge is employed as $M$, it leverages its broad understanding to generate synthetic data that is less reliant on the specific instances in $S_\text{anchor}$.
> - as the number of instances in the ICL setup increases, the LLM interprets this as a richer and more varied context, thereby increasing the output diversity instead.
> - a smaller HSIC value indicates a lower mutual information between the synthetic data and the training set, which leads to a larger information gain $\Delta I$.
> - with Theorem 4.7 and Theorem 4.10, this guarantees a tighter upper bound of the generalization error and higher GGMI, which contributes to the quality of synthetic data and increase the generalization capabilities.
>
> For $\Delta H$, we primarily focus on $H(S_{\text{gen}})$, and we adopt semantic entropy [10] as an estimator of the entropy of a dataset. We calculate the semantic entropy of the GPT-4o generated synthetic datasets and the results are as follows:
> $S_{\text{gen}}$ | 3-ins | 10-ins | 20-ins
> :--:|:--:|:--:|:--:
> **Semantic Entropy** | 1.0739 | 1.0503 | 1.0005
>
> **conclusion & analysis**
> - the semantic entropy of the synthetic data $S_{\text{gen}}$ is also negatively correlated with the number of instances in ICL.
> - increasing the sizes of $S_\text{anchor}$ when utilizing LLM as generative model $M$ can help reduce the entropy of the synthetic data.
> - reduction in entropy may be attributed to the richer and more varied context provided by a larger $S_\text{anchor}$, which enables $M$ to generate more accurate and informative synthetic data, thereby increasing the faithfulness of the synthetic data.
> - smaller semantic entropy indicates a lower entropy of the synthetic data $S_{\text{gen}}$, which leads to a larger $\Delta H$.
> - with Theorem 4.10, this benifts increasing the upper bound of GGMI, and contributes to the generalization capabilities of the model trained on the synthetic data.
>
> ## 5. Estimating on Different Models
> To further investigate the impact of different model architectures and parameters on the quality of synthetic data, we conduct experiments to evaluate the HSIC value and semantic entropy of the synthetic data $S_{\text{gen}}$ generated by different models.
> Due to computational resource limitations, we utilized GPT-4o, Llama-3.2-3B-Instruct, and Llama-3.2-1B-Instruct as the generative model $M$ to generate synthetic data with 3 instances in ICL setting.
>
> $M$ | GPT-4o | Llama-3.2-3B-Instruct | Llama-3.2-1B-Instruct
> :--:|:--:|:--:|:--:
> **HSIC** | 7.8703 | 11.4306 | /
> **Semantic Entropy** | 1.0739 | 2.9427 | /
>
> Note that under the prompt determined in the experimental setups, the Llama-3.2-1B-Instruct model did not adhere to the format requirements and failed to produce meaningful synthetic data. Consequently, the estimators are not available for this model. On the other hand, although Llama-3.2-3B-Instruct produced usable synthetic data, its quality was insufficient for fine-tuning GPT-2.
> **conclusion & analysis**
> - the generative model $M$ must possess sufficient instruction-following capabilities to generate synthetic data that can be effectively utilized to enhance model performance.
> - smaller model size of Llama-3.2-3B-Instruct compared to GPT-4o, results in a diminished capacity to generate synthetic data that is both faithful to and diverse from the anchor data.
>
> ## 6. Overall Conclusion
> **What Influence Synthetic Data Quality**
> - the quality of synthetic data is mainly reflected in two aspects: diversity and faithfulness.
> - diversity makes the synthetic data contain richer contents and thus increase the information gain $\Delta I$. With our theoretical analysis, this will benifit the generalization ability of the model post-trained on synthetic data.
> - Faithfulness makes the synthetic data semantically continuous, and thus decrease the entropy of the synthetic data $S_{\text{gen}}$, which also strenghten the generalization capabilities.
>
> **How to guarantee Quality in Practice**
> - in practice, the diversity and the faithfulness can be estimated by HSIC value and the semantic entropy, respectively, as demonstrated in the experimental settings of this section.
> - It is also important to highlight that employing a generative model with stronger instruction-following capabilities and more diverse knowledge can enhance the quality of synthetic data in both aspects.

---

> ### Author Response · Authors · 2024-11-20
> **Response Part 3**
>
> ## References
> [1] Mike Conover, Matt Hayes, Ankit Mathur, Jianwei Xie, Jun Wan, Sam Shah, Ali Ghodsi, Patrick Wendell, Matei Zaharia, and Reynold Xin. Free dolly: Introducing the world’s first truly open instruction-tuned llm, 2023. URL https://www.databricks.com/blog/2023/04/12/dolly-first-open-commercially-viable-instruction-tuned-llm.
>
> [2] Rohan Taori, Ishaan Gulrajani, Tianyi Zhang, Yann Dubois, Xuechen Li, Carlos Guestrin, Percy Liang, and Tatsunori B. Hashimoto. Stanford alpaca: An instruction-following llama model. https://github.com/tatsu-lab/stanford_alpaca, 2023.
>
> [3] OpenAI. Gpt-4o system card, 2024. URL https://arxiv.org/abs/2410.21276.
>
> [4] Meta. Introducing llama 3.2, 2024. URL https://www.llama.com/docs/model-cards-and-prompt-formats/llama3_2.
>
> [5] Alec Radford, Jeff Wu, Rewon Child, David Luan, Dario Amodei, and Ilya Sutskever. Language models are unsupervised multitask learners. 2019.
>
> [6] Lianmin Zheng, Wei-Lin Chiang, Ying Sheng, Siyuan Zhuang, Zhanghao Wu, Yonghao Zhuang, Zi Lin, Zhuohan Li, Dacheng Li, Eric Xing, et al. Judging llm-as-a-judge with mt-bench and chatbot arena. Advances in Neural Information Processing Systems, 36:46595–46623, 2023c.
>
> [7] Zhilin Wang, Alexander Bukharin, Olivier Delalleau, Daniel Egert, Gerald Shen, Jiaqi Zeng, Oleksii Kuchaiev, and Yi Dong. Helpsteer2-preference: Complementing ratings with preferences, 2024. URL https://arxiv.org/abs/2410.01257.
>
> [8] Chen Qian, Jie Zhang, Wei Yao, Dongrui Liu, Zhenfei Yin, Yu Qiao, Yong Liu, and Jing Shao. Towards tracing trustworthiness dynamics: Revisiting pre-training period of large language models. In Lun-Wei Ku, Andre Martins, and Vivek Srikumar (eds.), Findings of the Association for Computational Linguistics: ACL 2024, pp. 4864–4888, Bangkok, Thailand, August 2024. Association for Computational Linguistics. doi: 10.18653/v1/2024.findings-acl.290. URL https://aclanthology.org/2024.findings-acl.290.
>
> [9] Wan-Duo Kurt Ma, JP Lewis, and W Bastiaan Kleijn. The hsic bottleneck: Deep learning without back-propagation. In Proceedings of the AAAI conference on artificial intelligence, volume 34, pp. 5085–5092, 2020.
>
> [10] Sebastian Farquhar, Jannik Kossen, Lorenz Kuhn, and Yarin Gal. Detecting hallucinations in large language models using semantic entropy. Nature, 630(8017):625–630, 2024.
>
> # Empirical Experiments (Weakness 1 & 2)
> Thanks for your suggestions! We have included additional experiments on real-world LLM settings in Appendix G. You can also find a brief introduction above.
>
> # Analysis in Other Areas (Weakness 3)
> Thank you for your valuable advice!
>
> As our study focuses on the post-training stage, where the data size is significantly smaller than in the pre-training stage, we believe that the over-reliance phenomenon has a minimal impact. Moreover, the primary objective of our research is to understand why synthetic data is effective during post-training and to explore the underlying mechanisms, thereby providing guidance for synthetic data generation.
>
> That said, we fully agree that synthetic data can be studied from many other perspectives, such as understanding when it has a greater impact and how to model its relationship with specific downstream tasks. These are indeed fascinating and valuable directions for future research.
>
> # How would synthetic data that has not been processed by an LLM impact the proposed theory (Question 1)
> For non-LLM-produced synthetic data, we believe the crucial perspective still lies in modeling its distribution. In this paper, our primary focus is on modeling the distribution shift when using LLMs as the generative model. However, in cases such as rule-based synthetic data, we posit that these data also adhere to certain principles and follow specific distributions. This aligns with our theoretical findings and does not alter our conclusion: synthetic data should maintain both diversity and faithfulness to achieve higher quality.
>
> # Does this theoretical analysis still hold for non-text, multimodal data? (Question 2)
> Our results are derived from information theory, which provides the advantage of enabling general conclusions applicable across a wide range of scenarios. Specifically, we treat the data in our study as general training resources, without restricting it to text or sequence formats. As a result, the same principles can be extended to other data types without limitations.
>
> We hope our response addresses your questions and contributes to improving the quality of our study!
>
> Sincerely,
>
> The Authors

---

> > ### Comment · Reviewer_7KR4 · 2024-11-25
> >
> > Thank you for your detailed response to my comments. Your explanations have addressed my previous concerns. I will keep my overall rating but will increase the Soundness score from 2 to 3.

---

> ### Author Response · Authors · 2024-11-25
>
> Dear Reviewer 7KR4,
>
> I hope this message finds you well.
>
> We have carefully addressed the reviewers’ comments and made substantial revisions to our manuscript, as detailed in the responses and the revised PDF file. We have also summarized the key changes and enriched content to facilitate your review.
>
> Given that the review timeline is approaching its deadline, we kindly request your feedback on the revised submission at your earliest convenience. Your insights and comments are crucial for further improving the quality of our work, and we greatly value the opportunity for continued discussion.
>
> Thank you very much for your time and effort. Please do not hesitate to let us know if there are any additional clarifications or further details needed.
>
> Sincerely,
>
> The Authors

---

### Official Review · Reviewer_Re6N · 2024-11-03

**Soundness:** 3
**Presentation:** 4
**Contribution:** 4
**Rating:** 8
**Confidence:** 3

**Summary:**

This work provides a theoretical framework to analyse the nature and impact of synthetic data on the post-training process of LLMs. This work also proposes a "reverse-bottleneck" framework to visualize information flow in the synthetic generation process. On the basis of these theoretical principles, this work proceeds to mathematically derive useful upper bounds for model generalization when trained with synthetic data, in order to better quantify performance expectations.

**Strengths:**

- This work has a sound, well-researched theoretical basis
- The provided mathematical proofs are comprehensive and well-structured.
- The experimental setup is well-described and the theory is supported by some experimental results.

**Weaknesses:**

- This work could benefit from additional explanation of how diversity of generated data influences generalization capabilities, as well as a clearer explanation of the intuition behind how ∆I (the information gain derived from the model M) is associated with diversity (lines 483-485) and not just more unique but similar data points, possibly with examples.
-  The dimension of data on which these experiments are performed is 2; however, real-world data is often more complex and higher-dimensional. It would be illustrative to conduct the experiments documented in this paper on data of higher dimensions to reinforce that the theory holds for datasets of varying dimensionality.
- Additionally, given that this analysis is aimed at studying the behaviour of LLMs, the theoretical proofs presented in this work would benefit from conducting the same experiments either with LLMs or other non-GMM models to corroborate the theoretical inferences.

**Questions:**

- How does task difficulty affect generalization capability and the derived upper bounds? Are there any factors in the documented theoretical expressions that model or are associated with task complexity?
- Lemma 4.8: η' and W' are not defined prior to their usage in the expression; for clarity, it would be useful to define them as other variables have been.

---

> ### Author Response · Authors · 2024-11-20
> **Response Part 1**
>
> We would like to express our sincere gratitude for your valuable feedback and constructive suggestions on our paper. Thanks for your appreciation and advice!
>
> We noticed that all the reviewers suggest more experimental explorations on real-world settings, we would like to first introduce our enriched experiments, and then respond to your suggestions and questions.
>
> # Experiments on Real-World Settings
>  **The detailed presentation has been included in the Appendix G of our newly submitted PDF file.** We recommend referring to the paper for a thorough review. However, we still provide a brief summary of our experiments here (in the order of "generation", "training", and "evaluation").
>
> *Due to limitations in both training resources and time, the additional experiments may still be imperfect as a simulation of real-world LLMs. However, we have made every effort to present our analysis to the best of our ability. Training with LLMs is an extremely resource-consuming task, and we sincerely appreciate your understanding.*
>
> ## 0. Experimental Setup
> We follow the same setup as described in the main text, where the synthetic data $S_{\text{gen}}$ generated from a generative model $M$ prompted by $p$. We utilize a standard in-context learning (ICL) framework to determine $p$ using anchor data $S_{\text{anchor}}$, and we then evaluate the performance of the model trained on the synthetic data. Additionally, we estimate key components from our theoretical analysis in the main text, including information gain $\Delta I$ and the entropy of the synthetic data $H(S_{\text{gen}})$.
>
> ## 1. Generate Synthetic Data
> We apply Dolly-15K [1] as the benchmark dataset to sample $S_{\text{anchor}}$, then we modify the template from Alpaca [2] for determining the prompt $p$ for generating synthetic data. We adopt LLMs including GPT-4o [3], and latest Llama 3.2 models [4] like Llama-3.2-1B-Instruct and Llama-3.2-3B-Instruct as generative model $M$.
>
> By setting different numbers of instances of the ICL, we use GPT-4o to generate three datasets, noted as 3-ins, 10-ins and 20-ins respectively.
> GPT-4o Synthetic Dataset | 3-ins | 10-ins | 20-ins
> :------:|:---:|:---:|:---:
> **instance of ICL** | 3 | 10 | 20
>
> While for the two Llama 3.2 model, we generate one synthetic dataset respectively, in the setting of 3-ins.
>
> All the 5 synthetic datasets we generate are in the same size with benchmark datasets.
>
> ## 2. Fine-Tune with Synthetic Data
> We use the formerly GPT-4o generated synthetic datasets and the benchmark dataset to fine-tune a GPT-2 [5] model.
> The models are trained following a standard instruction-tuning framework, with fixed epochs to completely converge.
>
> ## 3. Evaluating the Quality of Synthetic Data
> We follow the framework of [6] and use Llama-3.1-Nemotron-70B-Instruct-HF [7] as the judge LLM to evaluate the fine-tuned models on the testing set of the benchmark dataset, and extract the ratings from its output. The final rating is averaged over the testing set to evaluate the performance of the fine-tuned model. After the nomalization, we use the ratings as a measure of the synthetic data quality.
> $S_{\text{gen}}$ | Base | 3-ins | 10-ins | 20-ins | real training set
> :----:|:------:|:-------:|:--------:|:--------:|:---:
> **Rating** | 0.1409 | 0.1836 | 0.1965 | 0.2015 | 0.2745
>
> The "Base" represents the unfine-tuned model.
>
> **conclusion & analysis**:
> - synthetic data quality is positively correlated with the number of instances in ICL.
> - appropriately increasing the number of instances in ICL can improve the quality of the synthetic data.
> - increasing the number of instances in the ICL prompts provides the generative model with a richer and more diverse context.
> - enhanced context allows the model to capture a broader range of patterns present in the anchor data, thereby generating synthetic data with richer content.
>
> ## 4. Estimate Components in Our Bound
> We primarily estimate the $\Delta I$ and $\Delta H$ in our bounds.
>
> For the information gain $\Delta I$, we follow [8] and [9] and use HSIC as an estimator of the mutual information between $S_{\text{gen}}$ and the benchmark training set. A larger HSIC indicates stronger relevance, and thus less information gain $\Delta I$. We calculate the HSIC value ($\times 10^{-3}$) of the GPT-4o generated synthetic datasets and the results are as follows:
> $S_{\text{gen}}$ | 3-ins | 10-ins | 20-ins
> :--:|:--:|:--:|:--:
> **HSIC** | 7.8703 | 7.8668 | 7.8502

---

> ### Author Response · Authors · 2024-11-20
> **Response Part 2**
>
> **conclusion & analysis**
> - more instances doesn't increase the HSIC value, but even lead to a lower HSIC value, indicating reduced mutual information between the synthetic data and the training set.
> - enlarging the sizes of $S_\text{anchor}$ does not significantly increase the dependency between the synthetic data and the training set, and may even enhance the diversity of the synthetic data.
> - when a LLM with a wide range of knowledge is employed as $M$, it leverages its broad understanding to generate synthetic data that is less reliant on the specific instances in $S_\text{anchor}$.
> - as the number of instances in the ICL setup increases, the LLM interprets this as a richer and more varied context, thereby increasing the output diversity instead.
> - a smaller HSIC value indicates a lower mutual information between the synthetic data and the training set, which leads to a larger information gain $\Delta I$.
> - with Theorem 4.7 and Theorem 4.10, this guarantees a tighter upper bound of the generalization error and higher GGMI, which contributes to the quality of synthetic data and increase the generalization capabilities.
>
> For $\Delta H$, we primarily focus on $H(S_{\text{gen}})$, and we adopt semantic entropy [10] as an estimator of the entropy of a dataset. We calculate the semantic entropy of the GPT-4o generated synthetic datasets and the results are as follows:
> $S_{\text{gen}}$ | 3-ins | 10-ins | 20-ins
> :--:|:--:|:--:|:--:
> **Semantic Entropy** | 1.0739 | 1.0503 | 1.0005
>
> **conclusion & analysis**
> - the semantic entropy of the synthetic data $S_{\text{gen}}$ is also negatively correlated with the number of instances in ICL.
> - increasing the sizes of $S_\text{anchor}$ when utilizing LLM as generative model $M$ can help reduce the entropy of the synthetic data.
> - reduction in entropy may be attributed to the richer and more varied context provided by a larger $S_\text{anchor}$, which enables $M$ to generate more accurate and informative synthetic data, thereby increasing the faithfulness of the synthetic data.
> - smaller semantic entropy indicates a lower entropy of the synthetic data $S_{\text{gen}}$, which leads to a larger $\Delta H$.
> - with Theorem 4.10, this benifts increasing the upper bound of GGMI, and contributes to the generalization capabilities of the model trained on the synthetic data.
>
> ## 5. Estimating on Different Models
> To further investigate the impact of different model architectures and parameters on the quality of synthetic data, we conduct experiments to evaluate the HSIC value and semantic entropy of the synthetic data $S_{\text{gen}}$ generated by different models.
> Due to computational resource limitations, we utilized GPT-4o, Llama-3.2-3B-Instruct, and Llama-3.2-1B-Instruct as the generative model $M$ to generate synthetic data with 3 instances in ICL setting.
>
> $M$ | GPT-4o | Llama-3.2-3B-Instruct | Llama-3.2-1B-Instruct
> :--:|:--:|:--:|:--:
> **HSIC** | 7.8703 | 11.4306 | /
> **Semantic Entropy** | 1.0739 | 2.9427 | /
>
> Note that under the prompt determined in the experimental setups, the Llama-3.2-1B-Instruct model did not adhere to the format requirements and failed to produce meaningful synthetic data. Consequently, the estimators are not available for this model. On the other hand, although Llama-3.2-3B-Instruct produced usable synthetic data, its quality was insufficient for fine-tuning GPT-2.
> **conclusion & analysis**
> - the generative model $M$ must possess sufficient instruction-following capabilities to generate synthetic data that can be effectively utilized to enhance model performance.
> - smaller model size of Llama-3.2-3B-Instruct compared to GPT-4o, results in a diminished capacity to generate synthetic data that is both faithful to and diverse from the anchor data.
>
> ## 6. Overall Conclusion
> **What Influence Synthetic Data Quality**
> - the quality of synthetic data is mainly reflected in two aspects: diversity and faithfulness.
> - diversity makes the synthetic data contain richer contents and thus increase the information gain $\Delta I$. With our theoretical analysis, this will benifit the generalization ability of the model post-trained on synthetic data.
> - Faithfulness makes the synthetic data semantically continuous, and thus decrease the entropy of the synthetic data $S_{\text{gen}}$, which also strenghten the generalization capabilities.
>
> **How to guarantee Quality in Practice**
> - in practice, the diversity and the faithfulness can be estimated by HSIC value and the semantic entropy, respectively, as demonstrated in the experimental settings of this section.
> - It is also important to highlight that employing a generative model with stronger instruction-following capabilities and more diverse knowledge can enhance the quality of synthetic data in both aspects.

---

> ### Author Response · Authors · 2024-11-20
> **Response Part 3**
>
> ## References
> [1] Mike Conover, Matt Hayes, Ankit Mathur, Jianwei Xie, Jun Wan, Sam Shah, Ali Ghodsi, Patrick Wendell, Matei Zaharia, and Reynold Xin. Free dolly: Introducing the world’s first truly open instruction-tuned llm, 2023. URL https://www.databricks.com/blog/2023/04/12/dolly-first-open-commercially-viable-instruction-tuned-llm.
>
> [2] Rohan Taori, Ishaan Gulrajani, Tianyi Zhang, Yann Dubois, Xuechen Li, Carlos Guestrin, Percy Liang, and Tatsunori B. Hashimoto. Stanford alpaca: An instruction-following llama model. https://github.com/tatsu-lab/stanford_alpaca, 2023.
>
> [3] OpenAI. Gpt-4o system card, 2024. URL https://arxiv.org/abs/2410.21276.
>
> [4] Meta. Introducing llama 3.2, 2024. URL https://www.llama.com/docs/model-cards-and-prompt-formats/llama3_2.
>
> [5] Alec Radford, Jeff Wu, Rewon Child, David Luan, Dario Amodei, and Ilya Sutskever. Language models are unsupervised multitask learners. 2019.
>
> [6] Lianmin Zheng, Wei-Lin Chiang, Ying Sheng, Siyuan Zhuang, Zhanghao Wu, Yonghao Zhuang, Zi Lin, Zhuohan Li, Dacheng Li, Eric Xing, et al. Judging llm-as-a-judge with mt-bench and chatbot arena. Advances in Neural Information Processing Systems, 36:46595–46623, 2023c.
>
> [7] Zhilin Wang, Alexander Bukharin, Olivier Delalleau, Daniel Egert, Gerald Shen, Jiaqi Zeng, Oleksii Kuchaiev, and Yi Dong. Helpsteer2-preference: Complementing ratings with preferences, 2024. URL https://arxiv.org/abs/2410.01257.
>
> [8] Chen Qian, Jie Zhang, Wei Yao, Dongrui Liu, Zhenfei Yin, Yu Qiao, Yong Liu, and Jing Shao. Towards tracing trustworthiness dynamics: Revisiting pre-training period of large language models. In Lun-Wei Ku, Andre Martins, and Vivek Srikumar (eds.), Findings of the Association for Computational Linguistics: ACL 2024, pp. 4864–4888, Bangkok, Thailand, August 2024. Association for Computational Linguistics. doi: 10.18653/v1/2024.findings-acl.290. URL https://aclanthology.org/2024.findings-acl.290.
>
> [9] Wan-Duo Kurt Ma, JP Lewis, and W Bastiaan Kleijn. The hsic bottleneck: Deep learning without back-propagation. In Proceedings of the AAAI conference on artificial intelligence, volume 34, pp. 5085–5092, 2020.
>
> [10] Sebastian Farquhar, Jannik Kossen, Lorenz Kuhn, and Yarin Gal. Detecting hallucinations in large language models using semantic entropy. Nature, 630(8017):625–630, 2024.
>
> # How diversity of generated data influences generalization capabilities (Weakness 1)
> We have conducted experiments to explore the influence of $\Delta I$ on the diversity of synthetic data. The intuition behind $\Delta I$ is that unique and diverse data points provide the model with more information during post-training, thereby enhancing generalization. This is why we refer to it as "information gain".
>
> For better explanation, Appendix G.4 includes a case study illustrating the detrimental effects of low diversity. Diverse data points enrich both form and content, thereby improving the effectiveness of LLM fine-tuning.
>
> # More Complex Experiments (Weakness 2 & 3)
> Thank you for your valuable suggestions! We have conducted experiments in the LLM settings, which operate in higher dimensions than before. The details of these experiments can be found in Appendix G or the sections above.
>
> # How does task difficulty affect generalization capability and the derived upper bounds (Question 1)
> As a fundamental result in information theory, the generalization error is upper-bounded by the mutual information between the training data $S$ and the model parameters $W$, i.e., $I(S, W)$. This mutual information term inherently reflects the task difficulty. Specifically, when a task is more challenging, the model must rely more heavily on the training data, leading to an increase in $I(S, W)$ and, consequently, a looser upper bound on the generalization error.
>
> We believe it would be valuable to incorporate more specific terms related to task difficulty into the bounds in future work. One potential direction is to measure the data structure as suggested in [11].
>
> # Not Defined Symbols (Question 2)
> $\eta^{'}$ and $W^{'}$ are the variables of the model trained with $S_\text{anchor}$, noted different from that of model trained with $S_\text{gen}$. $\eta^{'}$ is a constant depending on the information loss and $W^{'}$ is the model parameters.
> Thanks for your careful suggestions, we have added the definition in Lemma 4.8.
>
>
> [11] Steinke, T., & Zakynthinou, L. (2020, July). Reasoning about generalization via conditional mutual information. In Conference on Learning Theory (pp. 3437-3452). PMLR.
>
>
> We hope our response addresses your questions and contributes to improving the quality of our study!
>
> Sincerely,
>
> The Authors

---

> ### Author Response · Authors · 2024-11-25
>
> Dear Reviewer Re6N,
>
> I hope this message finds you well.
>
> We have carefully addressed the reviewers’ comments and made substantial revisions to our manuscript, as detailed in the responses and the revised PDF file. We have also summarized the key changes and enriched content to facilitate your review.
>
> Given that the review timeline is approaching its deadline, we kindly request your feedback on the revised submission at your earliest convenience. Your insights and comments are crucial for further improving the quality of our work, and we greatly value the opportunity for continued discussion.
>
> Thank you very much for your time and effort. Please do not hesitate to let us know if there are any additional clarifications or further details needed.
>
> Sincerely,
>
> The Authors

---

> > ### Comment · Reviewer_Re6N · 2024-11-27
> >
> > Thank you for your response, it helps clarify and add depth to the work performed. I would like to maintain my original score with an acknowledgement of the questions answered.

---

### Official Review · Reviewer_sWKh · 2024-11-04

**Soundness:** 3
**Presentation:** 3
**Contribution:** 3
**Rating:** 8
**Confidence:** 4

**Summary:**

The paper does a good job of establishing a theoretical rationale behind using synthetic data for LLM post-training processes like SFT. The modeling of synthetic data, focusing on distributional aspects is elaborate and properly justified as well. It establishes an intuitive information-theoretic based upper bound on the generalization error for an under-aligned LLM fine-tuned on synthetic data, and also delineates how this upper bound compares to the one when the LLM is tuned on just the real world samples (anchor data) - showcasing how the former proves to be a more stringent upper bound, leading to better generalization capabilities. Definitely helps bridge a gaping gap in LLM research!

**Strengths:**

1. The theoretical modeling of synthetic data and how it interacts with the output distribution of the LLM is intuitive and complete. Great job there!
2. The math behind generalization error bounds is solid.
3. The connection to the classical information bottleneck theory is novel and very intuitive, clearly justifies why synthetic data helps better performance on downstream task generalization.

**Weaknesses:**

1. The paper presents good theoretical foundations of "why" it is important to use synthetic data in post-training alignment and "how" it aids better generalization capabilities of the model. But it fails to address some really important "hows" that it promises in the Introduction section :
   - How does this study help in developing tailored synthetic data generation - that more effectively address specific gaps in training data, thereby enhancing the overall performance and generalization capabilities of large language models
   - How does this translate to better post-training practices involving synthetic datasets, from a practical perspective.
2. The assumption that the transformation function φT is reversible (Section 3.2) is unclear for real LLM prompting scenarios, as task-to-prompt relationships are often many-to-many.A better delineation of this point would be helpful.
3. ⁠The relationship between "synthetic factors" and actual LLM generation processes is not well defined.
4. The connection between the theoretical "information gain" concept and practical improvements in LLM performance is not clearly established.
5. I think its very crucial to address how the "diversity" and "quality" in synthetic data samples are a big part of why task generalization actually works when models are post trained on synthetic data. These should be adequately quantified and incorporated in the upper bound equations as well, for a more complete picture.

**Questions:**

Here are suggested experiments that could help validate the paper's hypotheses and theoretical frameworks:
1. LLM-based Validation Experiments :
     - ⁠Compare different sizes of anchor data and their corresponding synthetic data generation
     - ⁠Measure the relationship between model size and synthetic data quality
     - ⁠Test various prompting strategies and their impact on synthetic data diversity
     - Track information gain across different LLM architectures
2. Practical Application Questions:
    - How can practitioners use your theoretical bounds to improve their synthetic data generation process?
    - What specific guidance would you give for prompt engineering based on your theoretical findings?
3. Validation Questions:
    - How would your bounds change with different LLM architectures or sizes?
    -  ⁠What metrics would you recommend for measuring the quality of synthetic data in practice?

---

> ### Author Response · Authors · 2024-11-20
> **Response Part 1**
>
> We would like to express our sincere gratitude for your valuable feedback and constructive suggestions on our paper. Thanks for your appreciation and advice!
>
> We noticed that all the reviewers suggest more experimental explorations on real-world settings, we would like to first introduce our enriched experiments, and then respond to your suggestions and questions.
>
> # Experiments on Real-World Settings
> **The detailed presentation has been included in the Appendix G of our newly submitted PDF file.** We recommend referring to the paper for a thorough review. However, we still provide a brief summary of our experiments here (in the order of "generation", "training", and "evaluation").
>
> *Due to limitations in both training resources and time, the additional experiments may still be imperfect as a simulation of real-world LLMs. However, we have made every effort to present our analysis to the best of our ability. Training with LLMs is an extremely resource-consuming task, and we sincerely appreciate your understanding.*
>
> ## 0. Experimental Setup
> We follow the same setup as described in the main text, where the synthetic data $S_{\text{gen}}$ generated from a generative model $M$ prompted by $p$. We utilize a standard in-context learning (ICL) framework to determine $p$ using anchor data $S_{\text{anchor}}$, and we then evaluate the performance of the model trained on the synthetic data. Additionally, we estimate key components from our theoretical analysis in the main text, including information gain $\Delta I$ and the entropy of the synthetic data $H(S_{\text{gen}})$.
>
> ## 1. Generate Synthetic Data
> We apply Dolly-15K [1] as the benchmark dataset to sample $S_{\text{anchor}}$, then we modify the template from Alpaca [2] for determining the prompt $p$ for generating synthetic data. We adopt LLMs including GPT-4o [3], and latest Llama 3.2 models [4] like Llama-3.2-1B-Instruct and Llama-3.2-3B-Instruct as generative model $M$.
>
> By setting different numbers of instances of the ICL, we use GPT-4o to generate three datasets, noted as 3-ins, 10-ins and 20-ins respectively.
> GPT-4o Synthetic Dataset | 3-ins | 10-ins | 20-ins
> :------:|:---:|:---:|:---:
> **instance of ICL** | 3 | 10 | 20
>
> While for the two Llama 3.2 model, we generate one synthetic dataset respectively, in the setting of 3-ins.
>
> All the 5 synthetic datasets we generate are in the same size with benchmark datasets.
>
> ## 2. Fine-Tune with Synthetic Data
> We use the formerly GPT-4o generated synthetic datasets and the benchmark dataset to fine-tune a GPT-2 [5] model.
> The models are trained following a standard instruction-tuning framework, with fixed epochs to completely converge.
>
> ## 3. Evaluating the Quality of Synthetic Data
> We follow the framework of [6] and use Llama-3.1-Nemotron-70B-Instruct-HF [7] as the judge LLM to evaluate the fine-tuned models on the testing set of the benchmark dataset, and extract the ratings from its output. The final rating is averaged over the testing set to evaluate the performance of the fine-tuned model. After the nomalization, we use the ratings as a measure of the synthetic data quality.
> $S_{\text{gen}}$ | Base | 3-ins | 10-ins | 20-ins | real training set
> :----:|:------:|:-------:|:--------:|:--------:|:---:
> **Rating** | 0.1409 | 0.1836 | 0.1965 | 0.2015 | 0.2745
>
> The "Base" represents the unfine-tuned model.
>
> **conclusion & analysis**:
> - synthetic data quality is positively correlated with the number of instances in ICL.
> - appropriately increasing the number of instances in ICL can improve the quality of the synthetic data.
> - increasing the number of instances in the ICL prompts provides the generative model with a richer and more diverse context.
> - enhanced context allows the model to capture a broader range of patterns present in the anchor data, thereby generating synthetic data with richer content.
>
>
> ## 4. Estimate Components in Our Bound
> We primarily estimate the $\Delta I$ and $\Delta H$ in our bounds.
>
> For the information gain $\Delta I$, we follow [8] and [9] and use HSIC as an estimator of the mutual information between $S_{\text{gen}}$ and the benchmark training set. A larger HSIC indicates stronger relevance, and thus less information gain $\Delta I$. We calculate the HSIC value ($\times 10^{-3}$) of the GPT-4o generated synthetic datasets and the results are as follows:
> $S_{\text{gen}}$ | 3-ins | 10-ins | 20-ins
> :--:|:--:|:--:|:--:
> **HSIC** | 7.8703 | 7.8668 | 7.8502

---

> ### Author Response · Authors · 2024-11-20
> **Response Part 2**
>
> **conclusion & analysis**
> - more instances doesn't increase the HSIC value, but even lead to a lower HSIC value, indicating reduced mutual information between the synthetic data and the training set.
> - enlarging the sizes of $S_\text{anchor}$ does not significantly increase the dependency between the synthetic data and the training set, and may even enhance the diversity of the synthetic data.
> - when a LLM with a wide range of knowledge is employed as $M$, it leverages its broad understanding to generate synthetic data that is less reliant on the specific instances in $S_\text{anchor}$.
> - as the number of instances in the ICL setup increases, the LLM interprets this as a richer and more varied context, thereby increasing the output diversity instead.
> - a smaller HSIC value indicates a lower mutual information between the synthetic data and the training set, which leads to a larger information gain $\Delta I$.
> - with Theorem 4.7 and Theorem 4.10, this guarantees a tighter upper bound of the generalization error and higher GGMI, which contributes to the quality of synthetic data and increase the generalization capabilities.
>
> For $\Delta H$, we primarily focus on $H(S_{\text{gen}})$, and we adopt semantic entropy [10] as an estimator of the entropy of a dataset. We calculate the semantic entropy of the GPT-4o generated synthetic datasets and the results are as follows:
> $S_{\text{gen}}$ | 3-ins | 10-ins | 20-ins
> :--:|:--:|:--:|:--:
> **Semantic Entropy** | 1.0739 | 1.0503 | 1.0005
>
> **conclusion & analysis**
> - the semantic entropy of the synthetic data $S_{\text{gen}}$ is also negatively correlated with the number of instances in ICL.
> - increasing the sizes of $S_\text{anchor}$ when utilizing LLM as generative model $M$ can help reduce the entropy of the synthetic data.
> - reduction in entropy may be attributed to the richer and more varied context provided by a larger $S_\text{anchor}$, which enables $M$ to generate more accurate and informative synthetic data, thereby increasing the faithfulness of the synthetic data.
> - smaller semantic entropy indicates a lower entropy of the synthetic data $S_{\text{gen}}$, which leads to a larger $\Delta H$.
> - with Theorem 4.10, this benifts increasing the upper bound of GGMI, and contributes to the generalization capabilities of the model trained on the synthetic data.
>
> ## 5. Estimating on Different Models
> To further investigate the impact of different model architectures and parameters on the quality of synthetic data, we conduct experiments to evaluate the HSIC value and semantic entropy of the synthetic data $S_{\text{gen}}$ generated by different models.
> Due to computational resource limitations, we utilized GPT-4o, Llama-3.2-3B-Instruct, and Llama-3.2-1B-Instruct as the generative model $M$ to generate synthetic data with 3 instances in ICL setting.
>
> $M$ | GPT-4o | Llama-3.2-3B-Instruct | Llama-3.2-1B-Instruct
> :--:|:--:|:--:|:--:
> **HSIC** | 7.8703 | 11.4306 | /
> **Semantic Entropy** | 1.0739 | 2.9427 | /
>
> Note that under the prompt determined in the experimental setups, the Llama-3.2-1B-Instruct model did not adhere to the format requirements and failed to produce meaningful synthetic data. Consequently, the estimators are not available for this model. On the other hand, although Llama-3.2-3B-Instruct produced usable synthetic data, its quality was insufficient for fine-tuning GPT-2.
> **conclusion & analysis**
> - the generative model $M$ must possess sufficient instruction-following capabilities to generate synthetic data that can be effectively utilized to enhance model performance.
> - smaller model size of Llama-3.2-3B-Instruct compared to GPT-4o, results in a diminished capacity to generate synthetic data that is both faithful to and diverse from the anchor data.
>
> ## 6. Overall Conclusion
> **What Influence Synthetic Data Quality**
> - the quality of synthetic data is mainly reflected in two aspects: diversity and faithfulness.
> - diversity makes the synthetic data contain richer contents and thus increase the information gain $\Delta I$. With our theoretical analysis, this will benifit the generalization ability of the model post-trained on synthetic data.
> - Faithfulness makes the synthetic data semantically continuous, and thus decrease the entropy of the synthetic data $S_{\text{gen}}$, which also strenghten the generalization capabilities.
>
> **How to guarantee Quality in Practice**
> - in practice, the diversity and the faithfulness can be estimated by HSIC value and the semantic entropy, respectively, as demonstrated in the experimental settings of this section.
> - It is also important to highlight that employing a generative model with stronger instruction-following capabilities and more diverse knowledge can enhance the quality of synthetic data in both aspects.

---

> ### Author Response · Authors · 2024-11-20
> **Response Part 3**
>
> ## References
> [1] Mike Conover, Matt Hayes, Ankit Mathur, Jianwei Xie, Jun Wan, Sam Shah, Ali Ghodsi, Patrick Wendell, Matei Zaharia, and Reynold Xin. Free dolly: Introducing the world’s first truly open instruction-tuned llm, 2023. URL https://www.databricks.com/blog/2023/04/12/dolly-first-open-commercially-viable-instruction-tuned-llm.
>
> [2] Rohan Taori, Ishaan Gulrajani, Tianyi Zhang, Yann Dubois, Xuechen Li, Carlos Guestrin, Percy Liang, and Tatsunori B. Hashimoto. Stanford alpaca: An instruction-following llama model. https://github.com/tatsu-lab/stanford_alpaca, 2023.
>
> [3] OpenAI. Gpt-4o system card, 2024. URL https://arxiv.org/abs/2410.21276.
>
> [4] Meta. Introducing llama 3.2, 2024. URL https://www.llama.com/docs/model-cards-and-prompt-formats/llama3_2.
>
> [5] Alec Radford, Jeff Wu, Rewon Child, David Luan, Dario Amodei, and Ilya Sutskever. Language models are unsupervised multitask learners. 2019.
>
> [6] Lianmin Zheng, Wei-Lin Chiang, Ying Sheng, Siyuan Zhuang, Zhanghao Wu, Yonghao Zhuang, Zi Lin, Zhuohan Li, Dacheng Li, Eric Xing, et al. Judging llm-as-a-judge with mt-bench and chatbot arena. Advances in Neural Information Processing Systems, 36:46595–46623, 2023c.
>
> [7] Zhilin Wang, Alexander Bukharin, Olivier Delalleau, Daniel Egert, Gerald Shen, Jiaqi Zeng, Oleksii Kuchaiev, and Yi Dong. Helpsteer2-preference: Complementing ratings with preferences, 2024. URL https://arxiv.org/abs/2410.01257.
>
> [8] Chen Qian, Jie Zhang, Wei Yao, Dongrui Liu, Zhenfei Yin, Yu Qiao, Yong Liu, and Jing Shao. Towards tracing trustworthiness dynamics: Revisiting pre-training period of large language models. In Lun-Wei Ku, Andre Martins, and Vivek Srikumar (eds.), Findings of the Association for Computational Linguistics: ACL 2024, pp. 4864–4888, Bangkok, Thailand, August 2024. Association for Computational Linguistics. doi: 10.18653/v1/2024.findings-acl.290. URL https://aclanthology.org/2024.findings-acl.290.
>
> [9] Wan-Duo Kurt Ma, JP Lewis, and W Bastiaan Kleijn. The hsic bottleneck: Deep learning without back-propagation. In Proceedings of the AAAI conference on artificial intelligence, volume 34, pp. 5085–5092, 2020.
>
> [10] Sebastian Farquhar, Jannik Kossen, Lorenz Kuhn, and Yarin Gal. Detecting hallucinations in large language models using semantic entropy. Nature, 630(8017):625–630, 2024.
>
> # Improtant Hows (Weakness 1)
> We believe that our newly included experiments can partially help our study address the "hows" you mentioned.
> - To develop tailored synthetic data generation, one should focus on improving the quality of synthetic data. By the conclusion in Appendix G.5, it is crutial to guarantee the diversity and faithfulness of synthetic data.
> - In practice, one can utilize multiple tools (including the estimators we introduce in the experiments) to measure the diversity and faithfulness.
> - It is also important to guarantee the capability of instruction-following of the generative model $M$.
>
> # Better Delineation of Transformation Function (Weakness 2)
> We made the assumption that $\phi_\mathcal{T}$ is reversible primarily based on common ICL settings, where the prompt $p$ is a template filled with fixed instances, thus we can extract $S_{\text{anchor}}$ from the prompt $p$. However, we have determined that this assumption does not impact our theoretical results, as it does not alter the information flow in the synthetic data generation process. To enhance the clarity of our study, we have removed this assumption in the newly submitted version.
>
> # The Relationship between Synthetic Factors and LLM generation Process (Weakness 3)
> The synthetic factors are abstract assumptions that help seperate the influence of the prompt $p$ and generative model $M$, which are regarded as two crutial components in LLM generation process, for further theoritical analysis.
> With the synthetic factors established, we posit that the synthetic data $S_\text{gen}$ is primarily governed by two distinct factors: $e_p$ and $e_M$, which are actually assumed random variables related to the prompt $p$ and the LLM $M$ respectively.
> To make the assumptions more clear, we have enriched the explaination about the synthetic factors in the main text.
>
> # Connection between Theories and Practice (Weakness 4)
> We believe our above-mentioned experiments can strenthen the connection between $\Delta I$ and the practical improvements in LLM performance. $\Delta I$ is defined as the extra information introduced in the synthetic data generation progress, and it can guarantee the diversity of synthetic data, thus enhancing the LLM performance.

---

> ### Author Response · Authors · 2024-11-20
> **Response Part 4**
>
> # Quantify Diversity and Quality (Weakness 5)
> Since our theories are rooted in information theory, it can be somewhat challenging to precisely quantify exact values. However, there are tools available for estimation, and the settings used in our newly included experiments provide one such approach. We recommend assessing the quality by estimating the diversity and faithfulness of synthetic datasets, tailored to the specific tasks and models.
>
> # LLM-Based Validation Experiments (Question 1)
> We have already conducted experiments based on LLM. For your 4 valuable suggestions, the corresponding experiments are presented in:
> - Compare different sizes of anchor data and their corresponding synthetic data generation
>     - Appendix G.2
> - ⁠Measure the relationship between model size and synthetic data quality
>     - Appendix G.4
> - ⁠Test various prompting strategies and their impact on synthetic data diversity
>     - Appendix G.2 & G.3
> - Track information gain across different LLM architectures
>     - Appendix G.3 & G.4
>
> # Practical Application Question (Question 2)
> - Q: How can practitioners use your theoretical bounds to improve their synthetic data generation process?
>     - A: As presented in Appendix G, practitioners can focus on the synthetic data quality to improve the synthetic data generation process. And the synthetic data quality can be measured from diversity and faithfulness aspects, which can be estimated by multiple tools for estimating the mutual information or entropy.
> - Q: What specific guidance would you give for prompt engineering based on your theoretical findings?
>     - A: When performing prompt engineering, it is crutial to guarantee the alignment between the prompt $p$ and the generative model $M$. Since the aim is to increase the quality of synthetic data, the prompt needs to be designed for a more diverse and faithful output of the generative model. Specifically, some classic tricks such as "role-playing" or forcing the model not to use repeated words may take effect, which are already implemented in many famous open-source LLM synthetic dataset.
>
> # Validation Question (Question 3)
> - Q: How would your bounds change with different LLM architectures or sizes?
>     - A: We characterize synthetic data generation using information theory. However, information theory serves as a relatively high-level theoretical tool when modeling specific model architectures or algorithms. This model-agnostic framework offers the advantage of enabling general conclusions that are applicable across a wide range of scenarios (e.g., different modalities, tasks, etc.). In the future, we believe it would be valuable to explore bounds that account for model architectures or sizes. A potential direction is to incorporate the hypothesis space of the model [11] as a constraint or an upper bound on mutual information.
> - Q: What metrics would you recommend for measuring the quality of synthetic data in practice?
>     - A: As the answer above, we recommend to measure the diversity and faithfulness of the synthetic data. Specifically, in our experiments, we use HSIC to measure $\Delta I$ for diversity, and semantic entropy to measure $H(S_\text{gen})$ for faithfulness.
>
> [11] Lotfi, S., Finzi, M., Kuang, Y., Rudner, T. G., Goldblum, M., & Wilson, A. G. (2023). Non-vacuous generalization bounds for large language models. arXiv preprint arXiv:2312.17173.
>
>
> We hope our response addresses your questions and contributes to improving the quality of our study!
>
> Sincerely,
>
> The Authors

---

> ### Author Response · Authors · 2024-11-25
>
> Dear Reviewer sWKh,
>
> I hope this message finds you well.
>
> We have carefully addressed the reviewers’ comments and made substantial revisions to our manuscript, as detailed in the responses and the revised PDF file. We have also summarized the key changes and enriched content to facilitate your review.
>
> Given that the review timeline is approaching its deadline, we kindly request your feedback on the revised submission at your earliest convenience. Your insights and comments are crucial for further improving the quality of our work, and we greatly value the opportunity for continued discussion.
>
> Thank you very much for your time and effort. Please do not hesitate to let us know if there are any additional clarifications or further details needed.
>
> Sincerely,
>
> The Authors

---

### Official Review · Reviewer_qaT5 · 2024-11-04

**Soundness:** 2
**Presentation:** 2
**Contribution:** 2
**Rating:** 3
**Confidence:** 3

**Summary:**

The paper introduces the framework of understanding the LLM synthetic data generation process, by incorporating the reverse-bottleneck perspective. Using a reverse-bottleneck perspective, the authors model the generation of synthetic data and propose an information-theoretic metric called Generalization Gain via Mutual Information (GGMI) to assess the generalization performance of post-trained models. It also reports the information gain and generalization bound from previous literature.

**Strengths:**

1. The motivation is easy to follow and the figure is well represented.

2. The equation derived from previous literature, such as Lemma 3.1 and 4.4 is clear.

3. The paper tries to inject the LLM synthetic generation perspective using previously derived framework, showing a potential direction to understanding the synthetic data.

**Weaknesses:**

1. The paper lacks the experimental support, which cannot convince the readers of their assumption and derivations.

2. The paper is not organized in a good shape for guiding the readers for understanding their contributions. Instead, it enumerates the equations of generalization errors and gains, but lacking sufficient experimental results to support this.

3. Some derivations may be challenging for readers unfamiliar with advanced information theory, which may requires providing backgrounds about this. Since the authors did not conduct real-world experiments, it is unsure whether this can generalize to real-world settings. The framework assumes ideal conditions for synthetic data and LLMs, which may not generalize well to complex real-world scenarios with noisy data or biases.

**Questions:**

See above

---

> ### Author Response · Authors · 2024-11-20
> **Response Part 1**
>
> # Please Reconsider the Contribution
> We would like to express our sincere gratitude for your valuable feedback and constructive suggestions on our paper.
>
> However, we believe there may be some misunderstanding regarding the core contribution of our work. As a theoretical study, the primary goal of this paper is to model existing synthetic data utilization and analyze the mechanisms behind it. While the experiments are conducted primarily to verify our theoretical findings.
>
> Just as reviewer sWKH sated, our paper *"does a good job of establishing a theoretical rationale behind using synthetic data for LLM post-training processes like SFT"*, and as reviewer Re6N mentioned, our experiment *"is well-described and the theory is supported by some experimental results"*. To the best of our knowledge, we are the first to systematically model the synthetic data generation process and further analyze the generalization capabilities through this modeling. We believe that our paper can significantly contribute to understanding the mechanisms of synthetic data generation and utilization, thus providing a theoretical foundation for the design of synthetic data generation techniques and the optimization of the post-training process.
>
> Based on the above, we respectfully request a reconsideration of the evaluation of our work. We believe that its contribution should be viewed from a more theoretical perspective.
>
> # Experiments on Real-World Settings (Weakness 1 & 2 & 3)
> We would like to improve our paper with the suggestions you give in the "Weakness".
> Following your suggestions on enriching our experiments in real-world settings to provide experimental support (Weakness 1 & 2 & 3), we have conducted a group of new experiments, aiming to further investigate the process of synthetic data generation in real-world settings, by evaluating the quality of synthetic data produced under different conditions and identifying the factors that contribute to its effectiveness in enhancing model performance.
>
> **The detailed presentation has been included in the Appendix G of our newly submitted PDF file.** We recommend referring to the paper for a thorough review. However, we still provide a brief summary of our experiments here (in the order of "generation", "training", and "evaluation").
>
> *Due to limitations in both training resources and time, the additional experiments may still be imperfect as a simulation of real-world LLMs. However, we have made every effort to present our analysis to the best of our ability. Training with LLMs is an extremely resource-consuming task, and we sincerely appreciate your understanding.*
>
> ## 0. Experimental Setup
> We follow the same setup as described in the main text, where the synthetic data $S_{\text{gen}}$ generated from a generative model $M$ prompted by $p$. We utilize a standard in-context learning (ICL) framework to determine $p$ using anchor data $S_{\text{anchor}}$, and we then evaluate the performance of the model trained on the synthetic data. Additionally, we estimate key components from our theoretical analysis in the main text, including information gain $\Delta I$ and the entropy of the synthetic data $H(S_{\text{gen}})$.
>
> ## 1. Generate Synthetic Data
> We apply Dolly-15K [1] as the benchmark dataset to sample $S_{\text{anchor}}$, then we modify the template from Alpaca [2] for determining the prompt $p$ for generating synthetic data. We adopt LLMs including GPT-4o [3], and latest Llama 3.2 models [4] like Llama-3.2-1B-Instruct and Llama-3.2-3B-Instruct as generative model $M$.
>
> By setting different numbers of instances of the ICL, we use GPT-4o to generate three datasets, noted as 3-ins, 10-ins and 20-ins respectively.
> GPT-4o Synthetic Dataset | 3-ins | 10-ins | 20-ins
> :------:|:---:|:---:|:---:
> **instance of ICL** | 3 | 10 | 20
>
> While for the two Llama 3.2 model, we generate one synthetic dataset respectively, in the setting of 3-ins.
>
> All the 5 synthetic datasets we generate are in the same size with benchmark datasets.
>
> ## 2. Fine-Tune with Synthetic Data
> We use the formerly GPT-4o generated synthetic datasets and the benchmark dataset to fine-tune a GPT-2 [5] model.
> The models are trained following a standard instruction-tuning framework, with fixed epochs to completely converge.
>
> ## 3. Evaluating the Quality of Synthetic Data
> We follow the framework of [6] and use Llama-3.1-Nemotron-70B-Instruct-HF [7] as the judge LLM to evaluate the fine-tuned models on the testing set of the benchmark dataset, and extract the ratings from its output. The final rating is averaged over the testing set to evaluate the performance of the fine-tuned model. After the nomalization, we use the ratings as a measure of the synthetic data quality.
> $S_{\text{gen}}$ | Base | 3-ins | 10-ins | 20-ins | real training set
> :----:|:------:|:-------:|:--------:|:--------:|:---:
> **Rating** | 0.1409 | 0.1836 | 0.1965 | 0.2015 | 0.2745
>
> The "Base" represents the unfine-tuned model.

---

> ### Author Response · Authors · 2024-11-20
> **Response Part 2**
>
> **conclusion & analysis**:
> - synthetic data quality is positively correlated with the number of instances in ICL.
> - appropriately increasing the number of instances in ICL can improve the quality of the synthetic data.
> - increasing the number of instances in the ICL prompts provides the generative model with a richer and more diverse context.
> - enhanced context allows the model to capture a broader range of patterns present in the anchor data, thereby generating synthetic data with richer content.
>
>
> ## 4. Estimate Components in Our Bound
> We primarily estimate the $\Delta I$ and $\Delta H$ in our bounds.
>
> For the information gain $\Delta I$, we follow [8] and [9] and use HSIC as an estimator of the mutual information between $S_{\text{gen}}$ and the benchmark training set. A larger HSIC indicates stronger relevance, and thus less information gain $\Delta I$. We calculate the HSIC value ($\times 10^{-3}$) of the GPT-4o generated synthetic datasets and the results are as follows:
> $S_{\text{gen}}$ | 3-ins | 10-ins | 20-ins
> :--:|:--:|:--:|:--:
> **HSIC** | 7.8703 | 7.8668 | 7.8502
>
> **conclusion & analysis**
> - more instances doesn't increase the HSIC value, but even lead to a lower HSIC value, indicating reduced mutual information between the synthetic data and the training set.
> - enlarging the sizes of $S_\text{anchor}$ does not significantly increase the dependency between the synthetic data and the training set, and may even enhance the diversity of the synthetic data.
> - when a LLM with a wide range of knowledge is employed as $M$, it leverages its broad understanding to generate synthetic data that is less reliant on the specific instances in $S_\text{anchor}$.
> - as the number of instances in the ICL setup increases, the LLM interprets this as a richer and more varied context, thereby increasing the output diversity instead.
> - a smaller HSIC value indicates a lower mutual information between the synthetic data and the training set, which leads to a larger information gain $\Delta I$.
> - with Theorem 4.7 and Theorem 4.10, this guarantees a tighter upper bound of the generalization error and higher GGMI, which contributes to the quality of synthetic data and increase the generalization capabilities.
>
> For $\Delta H$, we primarily focus on $H(S_{\text{gen}})$, and we adopt semantic entropy [10] as an estimator of the entropy of a dataset. We calculate the semantic entropy of the GPT-4o generated synthetic datasets and the results are as follows:
> $S_{\text{gen}}$ | 3-ins | 10-ins | 20-ins
> :--:|:--:|:--:|:--:
> **Semantic Entropy** | 1.0739 | 1.0503 | 1.0005
>
> **conclusion & analysis**
> - the semantic entropy of the synthetic data $S_{\text{gen}}$ is also negatively correlated with the number of instances in ICL.
> - increasing the sizes of $S_\text{anchor}$ when utilizing LLM as generative model $M$ can help reduce the entropy of the synthetic data.
> - reduction in entropy may be attributed to the richer and more varied context provided by a larger $S_\text{anchor}$, which enables $M$ to generate more accurate and informative synthetic data, thereby increasing the faithfulness of the synthetic data.
> - smaller semantic entropy indicates a lower entropy of the synthetic data $S_{\text{gen}}$, which leads to a larger $\Delta H$.
> - with Theorem 4.10, this benifts increasing the upper bound of GGMI, and contributes to the generalization capabilities of the model trained on the synthetic data.
>
> ## 5. Estimating on Different Models
> To further investigate the impact of different model architectures and parameters on the quality of synthetic data, we conduct experiments to evaluate the HSIC value and semantic entropy of the synthetic data $S_{\text{gen}}$ generated by different models.
> Due to computational resource limitations, we utilized GPT-4o, Llama-3.2-3B-Instruct, and Llama-3.2-1B-Instruct as the generative model $M$ to generate synthetic data with 3 instances in ICL setting.
>
> $M$ | GPT-4o | Llama-3.2-3B-Instruct | Llama-3.2-1B-Instruct
> :--:|:--:|:--:|:--:
> **HSIC** | 7.8703 | 11.4306 | /
> **Semantic Entropy** | 1.0739 | 2.9427 | /
>
> Note that under the prompt determined in the experimental setups, the Llama-3.2-1B-Instruct model did not adhere to the format requirements and failed to produce meaningful synthetic data. Consequently, the estimators are not available for this model. On the other hand, although Llama-3.2-3B-Instruct produced usable synthetic data, its quality was insufficient for fine-tuning GPT-2.
>
> **conclusion & analysis**
> - the generative model $M$ must possess sufficient instruction-following capabilities to generate synthetic data that can be effectively utilized to enhance model performance.
> - smaller model size of Llama-3.2-3B-Instruct compared to GPT-4o, results in a diminished capacity to generate synthetic data that is both faithful to and diverse from the anchor data.

---

> ### Author Response · Authors · 2024-11-20
> **Response Part 3**
>
> ## 6. Overall Conclusion
> **What Influence Synthetic Data Quality**
> - the quality of synthetic data is mainly reflected in two aspects: diversity and faithfulness.
> - diversity makes the synthetic data contain richer contents and thus increase the information gain $\Delta I$. With our theoretical analysis, this will benifit the generalization ability of the model post-trained on synthetic data.
> - Faithfulness makes the synthetic data semantically continuous, and thus decrease the entropy of the synthetic data $S_{\text{gen}}$, which also strenghten the generalization capabilities.
>
> **How to guarantee Quality in Practice**
> - in practice, the diversity and the faithfulness can be estimated by HSIC value and the semantic entropy, respectively, as demonstrated in the experimental settings of this section.
> - It is also important to highlight that employing a generative model with stronger instruction-following capabilities and more diverse knowledge can enhance the quality of synthetic data in both aspects.
>
> ## References
> [1] Mike Conover, Matt Hayes, Ankit Mathur, Jianwei Xie, Jun Wan, Sam Shah, Ali Ghodsi, Patrick Wendell, Matei Zaharia, and Reynold Xin. Free dolly: Introducing the world’s first truly open instruction-tuned llm, 2023. URL https://www.databricks.com/blog/2023/04/12/dolly-first-open-commercially-viable-instruction-tuned-llm.
>
> [2] Rohan Taori, Ishaan Gulrajani, Tianyi Zhang, Yann Dubois, Xuechen Li, Carlos Guestrin, Percy Liang, and Tatsunori B. Hashimoto. Stanford alpaca: An instruction-following llama model. https://github.com/tatsu-lab/stanford_alpaca, 2023.
>
> [3] OpenAI. Gpt-4o system card, 2024. URL https://arxiv.org/abs/2410.21276.
>
> [4] Meta. Introducing llama 3.2, 2024. URL https://www.llama.com/docs/model-cards-and-prompt-formats/llama3_2.
>
> [5] Alec Radford, Jeff Wu, Rewon Child, David Luan, Dario Amodei, and Ilya Sutskever. Language models are unsupervised multitask learners. 2019.
>
> [6] Lianmin Zheng, Wei-Lin Chiang, Ying Sheng, Siyuan Zhuang, Zhanghao Wu, Yonghao Zhuang, Zi Lin, Zhuohan Li, Dacheng Li, Eric Xing, et al. Judging llm-as-a-judge with mt-bench and chatbot arena. Advances in Neural Information Processing Systems, 36:46595–46623, 2023c.
>
> [7] Zhilin Wang, Alexander Bukharin, Olivier Delalleau, Daniel Egert, Gerald Shen, Jiaqi Zeng, Oleksii Kuchaiev, and Yi Dong. Helpsteer2-preference: Complementing ratings with preferences, 2024. URL https://arxiv.org/abs/2410.01257.
>
> [8] Chen Qian, Jie Zhang, Wei Yao, Dongrui Liu, Zhenfei Yin, Yu Qiao, Yong Liu, and Jing Shao. Towards tracing trustworthiness dynamics: Revisiting pre-training period of large language models. In Lun-Wei Ku, Andre Martins, and Vivek Srikumar (eds.), Findings of the Association for Computational Linguistics: ACL 2024, pp. 4864–4888, Bangkok, Thailand, August 2024. Association for Computational Linguistics. doi: 10.18653/v1/2024.findings-acl.290. URL https://aclanthology.org/2024.findings-acl.290.
>
> [9] Wan-Duo Kurt Ma, JP Lewis, and W Bastiaan Kleijn. The hsic bottleneck: Deep learning without back-propagation. In Proceedings of the AAAI conference on artificial intelligence, volume 34, pp. 5085–5092, 2020.
>
> [10] Sebastian Farquhar, Jannik Kossen, Lorenz Kuhn, and Yarin Gal. Detecting hallucinations in large language models using semantic entropy. Nature, 630(8017):625–630, 2024.
>
>
> # More Background about Information Theory (Weakness 3)
> In Weakness 3, you also suggested providing more background on information theory. We would like to clarify that we have already introduced the relevant concepts and definitions in Section 2.3 of the main text and Appendix A. To further address your suggestion, we have enriched Appendix A by formally introducing the concept of information bottleneck theory. We hope these efforts enhance the clarity and presentation of our study.
>
> There are also concerns regarding the idealized assumptions made in the paper. However, we believe that such assumptions are essential for generalizing to a broader range of scenarios in theoretical studies. While our results represent an abstract modeling of real-world algorithms, they offer valuable insights that can inform and guide practical engineering projects.
>
> Above all, we respectfully request your reconsideration and look forward to your response!
>
> Sincerely,
>
> The Authors

---

> ### Author Response · Authors · 2024-11-25
>
> Dear Reviewer qaT5,
>
> I hope this message finds you well.
>
> We have carefully addressed the reviewers’ comments and made substantial revisions to our manuscript, as detailed in the responses and the revised PDF file. We have also summarized the key changes and enriched content to facilitate your review.
>
> Given that the review timeline is approaching its deadline, we kindly request your feedback on the revised submission at your earliest convenience. Your insights and comments are crucial for further improving the quality of our work, and we greatly value the opportunity for continued discussion.
>
> Thank you very much for your time and effort. Please do not hesitate to let us know if there are any additional clarifications or further details needed.
>
> Sincerely,
>
> The Authors

---

> ### Comment · Reviewer_qaT5 · 2024-11-26
>
> Thank you for your detailed response. After carefully reviewing the author's rebuttal, there are **significant concerns** that remain unresolved:
>
> 1. The paper claims its main theoretical contributions lie in the derivation of e.g., **Lemma 3.1 and Lemma 4.1**. However, these results appear to be almost identical, to prior work in the literature, particularly [1] and [2]. **The authors should clearly tell how their theoretical contributions compared with these works and provide a detailed comparison**. Without this clarification, the novelty of the theoretical contribution remains questionable.
>
> References:
>
> [1] Toward Understanding Generative Data Augmentation. NeurIPS 2023. https://proceedings.neurips.cc/paper_files/paper/2023/file/a94a8800a4b0af45600bab91164849df-Paper-Conference.pdf
>
> [2] An Information-Theoretic View for Deep Learning. https://arxiv.org/pdf/1804.09060
>
> 2. The authors should clearly report the data size of synthetic data. Also,  in response 5, all results of "Llama-3.2-1B-Instruct" are missing.
>
> After carefully reviewing the author's response, I tend to keep my original score. At least the authors should fairly compare their claimed theoretical contributions with previous literatures [1][2].

---

> ### Author Response · Authors · 2024-11-26
> **Response (2nd round, part 1)**
>
> Thanks for your responses and further discussions!
> We are willing to address your two main concerns one by one.
>
> ## 1. About the Theoretical Contribution
> ***Before our response, there is not a "Lemma 4.1" in our paper, we guess you mean "Lemma 4.4"?***
>
> If you mean **Lemma 3.1** and **Lemma 4.4**, we are willing to clearly tell our theoretical contributions in our paper as follows.
>
> First, both in the abstract and the conclusion part, we claim that our two main contributions are:
> > 1. **We demonstrate that the generalization capability of the post-trained model is critically determined by the information gain derived from the generative model.**
> > 2. **We introduce the concept of Generalization Gain via Mutual Information (GGMI) and elucidate the relationship between generalization gain and information gain.**
>
> We hope we can reach an agreement on this: **The main contribution of our paper lies in the above two.**
>
>
> These two contributions correspond to **Theorem 4.7** and **Theorem 4.10** respectively. ***Actually, in this paper we only call these two conclusions "theorems".***
> While the **Lemma 3.1** and **Lemma 4.4** only serve as the foundation or starting point of our theoretical derivation.
>
> Based on the above agreement, we would like to sort out the origins and derivation processes of the two theorems respectively.
>
> ### About Theorem 4.7
> This conclusion is built on **Lemma 3.1** and **Lemma 4.4**, which is true.
> But we believe that the core contribution in deriving **Theorem 4.7** is actually Lemma 4.5, **which is completely original in our paper.**
>
> To be more specific, we would like to list the roles of all the related equations, definitions, and lemmas in deriving **Theorem 4.7**, in the order of our derivation.
>
> - **In Line 263, we define the generalization error of the LLM post-trained on $S_\text{gen}$.** This is the beginning of derivation. Also, the goal is to bound this generalization error.
> - **In Lemma 3.1, we decouple the generalization error**, and bound **part of it** (Distributions' Divergence) similar to [1].
> - **Core Contribution:** Bound the other part by $\Delta I$.
>     - ***[Core Contribution]* From Line 301 to Line 350, we are making necessary definitions.** By doing so, we are preparing to bound **the other part** (Generalization Error w.r.t. synthetic data) of the generalization error.
>     - **In Lemma 4.4**, it serves as a theoretical foundation proposed in [2], which links the generalization error with the mutual information item. **We take this as the premise for our derivation.**
>     - ***[Core Contribution]* In Lemma 4.5, we prove that the mutual information item in Lemma 4.4 can be further upper bounded by $\Delta I$**, where $\Delta I$ is the originally proposed by our paper.
>     - **In Lemma 4.6, we combine Lemma 4.4 and Lemma 4.5.** This finished bounding the other part, i.e., Generalization Error w.r.t. synthetic data.
> - **In Theorem 4.7, we combine Lemma 3.1 and Lemma 4.6.** This finished bounding all parts of the goal.
>
> To conclude, Lemma 3.1 and Lemma 4.4 are necessary theoretical foundations in deriving Theorem 4.7, which have been appropriately cited. We have marked our core contribution in the process above. We believe that it is not "identical" to previous works.
>
> ### About Theorem 4.10
> Different from Theorem 4.7, **Theorem 4.10** is derived directly from **Definition 4.9**. Since **Definition 4.9** is first proposed in our paper and serves as one of our main contributions, we believe that the results in **Theorem 4.10** have sufficient novelty.
>
>
> ## 2. About the Dataset Size
> Though we have introduced the synthetic data size in our first rebuttal response (in the last line in "1. Generate Synthetic Data") that:
> > All the 5 synthetic datasets we generate are the same size as benchmark datasets.
>
> We would like to clarify it more clearly. That is, **All the 5 synthetic datasets we generate contain 12,000 rows of data, which is the same size as the training set of the benchmark dataset**.
>
> ## 3. About the Missing Result of "Llama-3.2-1B-Instruct"
> Actually, they are not missing.
>
> **In our first rebuttal response** (in the second paragraph of "5. Estimating on Different Models"), we explained:
> > Note that under the prompt determined in the experimental setups, the Llama-3.2-1B-Instruct model did not adhere to the format requirements and failed to produce meaningful synthetic data. Consequently, the estimators are not available for this model.
>
> Also, **in the caption of Table 4**, we explained:
> > Note that the Llama-3.2-1B-Instruct model did not adhere to the format requirements and thus failed to produce meaningful synthetic data.
>
> And **in line 1240 of our paper**, we explained:
> > Note that under the prompt determined in the experimental setups, the Llama-3.2-1B-Instruct model did not adhere to the format requirements and failed to produce meaningful synthetic data.

---

> ### Author Response · Authors · 2024-11-26
> **Response (2nd round, part 2)**
>
> More specifically, since the prompt is too complex, Llama-3.2-1B-Instruct tends to just repeat the input prompts and cannot respond in a JSON style, so it is meaningless to calculate the corresponding results.
>
> The reason for maintaining the report of this phenomenon is also listed **in our paper line 1241**:
> > This observation underscores a fundamental premise that the generative model $M$ must possess sufficient instruction-following capabilities to generate synthetic data that can be effectively utilized to enhance model performance.
>
> **Though the generation failed, this phenomenon can still indicate this fundamental premise when generating synthetic data in practice.**
>
> We hope our explanation can help you better understand our theoretical contributions and address your concerns about the newly attached experiment setting.
>
> Sincerely,
>
> The Authors

---

> ### Comment · Reviewer_qaT5 · 2024-12-02
>
> Thank you for your response. The author states their main contributions are Theorem 4.7 and Theorem 4.10. It is clear that Theorem 4.7 is an injection of Eq. (15) [2] and Section 3.1 [1]. The authors spent a large number of wordings describing the previous framework in [1] and [2] and mixing their derivations and previous literature. The current paper is not in a good shape. It is required to clearly state their contribution on top of previous works.
>
> The proposed synthetic can be irrelevant to LLMs at all (can be any neural networks), considering that there is no specifics tailed to LLM scenarios. Besides, the authors leave blank at the column of "Llama-3.2-1B-Instruct", only mentioning that "Llama-3.2-1B-Instruct model did not adhere to the format requirements and failed to produce meaningful synthetic data." It is confusing of mentioning an LLMs that cannot be applied in the proposed methods and leave the results empty. More importantly, there are lots of LLMs that are SFT-ed or RLHF-ed, similar to "Llama-3.2-1B-Instruct". If "Llama-3.2-1B-Instruct" cannot be applied in such framework, it will take the same thing in such models. It is confusing purely mentioning "format requirements" without any explanations. Also, the HSIC metrics authors applied require a thorough introduction.
>
> Moreover, it is required to clearly state the details of experiments on synthetic data, especially of how does the data generated and filtering, how to select the prompt and balance the data diversity, etc. However, the only experiments reported in the main paper is a toy experiments on GMM simulation (Section 4.4). It is unknown about how the paper correlates with LLM synthetic data merely from the main paper.
>
> After carefully reading the paper, the paper still requires major revisions to meet the standard of ICLR. I am maintaining my overall score and increasing the soundness score.

---

> ### Author Response · Authors · 2024-12-03
> **Response (3rd round, part 1)**
>
> Thanks for your response. We regret that, despite multiple discussions, there still appear to remain a significant misunderstanding of our work.
>
> We would like to restate our contributions clearly again:
> **This study provides the first mathematical modeling of the role of synthetic data in the post-training of LLMs.** Following this modeling, we utilize **information theory** to offer a simple yet rigorous proof that **links the generalization capability of LLMs with information gain.** Given the critical role synthetic data plays in the post-training of LLMs, we believe our findings represent an important and valuable contribution to this field. **Does an article must necessarily require a lengthy and overly complicated proof process before you can call it a sufficient contribution?**
>
> If you disagree with our perspective, **we kindly request that you provide references to works you consider to have made sufficient contributions in the areas of synthetic data or information theory.**
>
> Next, we will explain your potential misunderstandings about our work.
>
> As you noted, our key contributions lie in Theorem 4.7 and Theorem 4.10. **We are puzzled by the continued focus on Theorem 4.7 alone, with little acknowledgment of Theorem 4.10.** We believe it is insufficient to evaluate our contributions solely based on **a subset of the theoretical results presented in the paper.**
>
> Regarding Theorem 4.7, **we are surprised that it is still perceived as "an injection of Eq. (15) [2] and Section 3.1 [1]"** despite our clarifications during the second round of discussions. We would like to reiterate that our **core contributions lie in characterizing the information gain ($\Delta I$)** and **bounding Generalization Error w.r.t. synthetic data in Lemma 4.5**, as we marked as **"[Core Contribution]"** in our second-round response.
>
> We also find it unreasonable to accuse us of "spending a large number of wordings describing the previous framework" or "mixing their derivations and previous literature". The references to previous work are limited to Lemma 3.1 (approximately 13 lines) and Lemma 4.4 (approximately 12 lines), amounting to 25 lines out of a total of 540 lines in the 10-page main text. **How can such a small portion of the paper be deemed "a large number of wordings"?** Furthermore, we have clearly cited the relevant literature on lines 266 and 354. **The assertion that we are "mixing derivations and previous literature" is similarly unfounded**.
>
> We actually feel confused about your standard about "good shape" for a theoritical paper. **Academic research inherently builds upon prior work, that is the nature of research.** and the cited references you frequently mention are also building upon previous researches, e.g., [1] is built upon [3] and [4], [2] is built on [5],[6] and [7]. Both [1] and [2] contribute meaningful findings despite building upon prior literature, **just as our study does.** Theories develop from previous theories, togerher with the practice. **If utilizing prior results as a foundation renders a theoretical paper “not in good shape,” we kindly request examples of theoretical works you consider “in good shape”.**
>
> You have also proposed the concern that our framework has no specifics tailored to LLM. We kindly refer you to Section 4.1, and also the Theorem 4.10 you ignore. **Numerous real-world synthetic datasets are generated using the paradigm outlined in this paper**. This paradigm forms the foundation of the reverse-bottleneck framework. **If you believe it is unrelated to LLMs, we invite you to cite references demonstrating the reverse-bottleneck effect without involving LLMs.**
>
> **We have explained more than 4 times about the results of Llama-3.2-1B-Instruct,** including in our first-round rebuttal, Table 4 caption, line 1240, and the second-round rebuttal. The key takeaway is not the format but the **lack of meaningful output due to the model’s repetition of the prompt**. This result highlights a key premise of the popular synthetic data generation paradigm: **the generative model $M$ must possess enough instruction-following capability.**

---

> ### Author Response · Authors · 2024-12-03
> **Response (3rd round, part 2)**
>
> As is common in theoretical research, **simulation experiments suffice to support main results**,  as demonstrated in [1] and other works across various domains [8], [9], [10]. While we are willing to conduct more extensive experiments, testing across a broad range of LLMs is resource-intensive and unfeasible within the limited review period. In our first-round rebuttal, we presented additional experiments to address reviewers’ concerns to the best of our ability. But we need to emphasize that, **the theorems in the main text are the crucial parts we want to present**. Due to constraints on page and time limits, **we prioritize presenting our theoretical findings in the main text, with experimental details provided in Appendix B and Appendix G**. These appendices include thorough descriptions of GMM simulation and LLM experiment settings, **covering generation procedures and prompts**.
>
> Finally, regarding your comment that this paper does not meet the ICLR standard:
>
> **Our research group has published over 30 papers in ICLR, NeurIPS, and ICML**. From our experience, **we strongly believe this paper meets the standards of ICLR and is of higher quality than many of our previous publications**.
>
> ## References
>
> [1] Zheng, C., Wu, G., & Li, C. (2023). Toward understanding generative data augmentation. Advances in neural information processing systems, 36, 54046-54060.
>
> [2] Zhang, J., Liu, T., & Tao, D. (2018). An information-theoretic view for deep learning. arXiv preprint arXiv:1804.09060.
>
> [3] Bousquet, O., Klochkov, Y., & Zhivotovskiy, N. (2020, July). Sharper bounds for uniformly stable algorithms. In Conference on Learning Theory (pp. 610-626). PMLR.
>
> [4] Stéphane Boucheron, Gábor Lugosi, and Pascal Massart. Concentration inequalities: A nonasymptotic theory of independence. Oxford university press, 2013.
>
> [5] Ahlswede, R. and Gács, P. (1976). Spreading of sets in product spaces and hypercontraction of the Markov operator. The annals of probability, pages 925–939.
>
> [6] Shwartz-Ziv, R. and Tishby, N. (2017). Opening the Black Box of Deep Neural Networks via Information. ArXiv e-prints.
>
> [7] Shalev-Shwartz, S., Shamir, O., Srebro, N., and Sridharan, K. (2010). Learnability, stability and uniform convergence. Journal of Machine Learning Research, 11(Oct):2635–2670.
>
> [8] Ben-Shaul, I., Shwartz-Ziv, R., Galanti, T., Dekel, S., & LeCun, Y. (2023). Reverse engineering self-supervised learning. Advances in Neural Information Processing Systems, 36, 58324-58345.
>
> [9] Huang, W., Yi, M., Zhao, X., & Jiang, Z. (2021). Towards the generalization of contrastive self-supervised learning. arXiv preprint arXiv:2111.00743.
>
> [10] Vasudeva, B., Deora, P., & Thrampoulidis, C. (2024). Implicit bias and fast convergence rates for self-attention. arXiv preprint arXiv:2402.05738.

---

### Author Response · Authors · 2024-11-25
**Summary of the Revised PDF File**

To facilitate your review, we provide a summary of the changes and enriched content in our newly submitted PDF, as per the reviewers’ advice.
1. **Enhanced Background on Information Theory.** To provide a more comprehensive background on information theory, we have enriched Appendix A.2 with a detailed illustration. We first introduce the foundational concepts and objectives of the information bottleneck method and then review the connections between generalization error and information theory. We hope this enhancement aids readers unfamiliar with information theory in better understanding our study.
2. **Improved Delineation of the Transformation Function.** Previously, we assumed that $\phi_\mathcal{T}$ is a reversible function, based on the common ICL setting where the prompt $p$ is a template filled with several instances. Upon further reflection, we realized this assumption is unnecessary as it does not affect the information flow in the synthetic data generation process, which is central to our analysis. To improve clarity and avoid potential misunderstandings, we have removed this assumption in Section 3.2.
3. **Clearer Illustration of Synthetic Factors.** Synthetic factors are defined to separate the influence of the prompt $p$ and the generative model $M$. To clarify this definition and emphasize its purpose, we have added a brief explanation of the underlying intuition in Definition 4.1.
4. **Better Defined Symbols.** Due to page limitations, we had previously omitted explanations for $\eta^{'}$ and $W^{'}$ in Lemma 4.8. To improve clarity, we now define these symbols explicitly as variables trained with $S_{\text{anchor}}$ rather than $S_{\text{gen}}$. This explanation has been added to Lemma 4.8.
5. **More Precise Explanation of Diversity and Faithfulness.** To better align with our empirical findings, we have revised the explanation of the two synthetic data generation principles, “diversity” and “faithfulness,” in Subsection 4.3, ensuring they are more directly connected to our theoretical results.
6. **Empirical Experiments on LLM Setting.** To strengthen the connection between our theoretical findings and practical applications, we conducted a series of experiments detailed in Appendix G. In this section, we address the following points:
    - We empirically demonstrate that language models post-trained on synthetic data with higher “diversity” and “faithfulness” generalize better on testing data, aligning with Theorems 4.7 and 4.10. (Appendix G.2)
    - We recommend practical measurements of “diversity” and “faithfulness” based on specific components from our theoretical findings, namely $\Delta I$ and $\Delta H$, respectively. (Appendix G.3)
    - We propose practical estimators for these principles. Specifically, we use HSIC to estimate the negative correlation with $\Delta I$ and semantic entropy to estimate $H(S_{\text{gen}})$, thus measuring $\Delta H$. (Appendix G.3)
    - We extend and validate our findings across different models and include a brief case study to illustrate the negative impact of low diversity. (Appendix G.4)

We hope this summary helps you review our newly added content with greater clarity!

---

### Meta-Review · Area_Chair_Aunc · 2024-12-20

**Metareview:**

The paper proposes a theoretical framework to analyze the role of synthetic data in the post-training of LLMs.

 *strengths*
- Solid theoretical foundations and well-structured mathematical proofs to support the claims.
- Novel concepts like the reverse-bottleneck framework and GGMI that provide valuable insights into the role of synthetic data.
- Good job in establishing the theoretical rationale for using synthetic data in LLM post-training.

*weaknesses*
- The lack of extensive empirical validation on real-world LLM settings, which raises concerns about the applicability of the theoretical findings.
- The focus on the performance benefits of synthetic data, without a thorough analysis of potential downsides or limitations.
- The need for more concrete guidance on how the theoretical insights can be leveraged in practical LLM post-training workflows.

Overall,  it is marginally above the acceptance threshold: the reviewers acknowledged the solid theoretical contributions of the paper, but felt that more work is needed to bridge the gap between the theory and real-world LLM applications.

**Additional Comments On Reviewer Discussion:**

During the rebuttal period, the authors provided a detailed response addressing the reviewers' concerns and feedback.

- They presented additional experiments conducted on real-world LLM settings.
- The authors also clarified their theoretical contributions, emphasizing that the core novelty lies in the characterization of information gain and the introduction of GGMI, rather than just deriving bounds from previous work.
- Additionally, they addressed specific questions regarding the relationship between the theoretical analysis and practical implementation, as well as the applicability of the framework to non-text and multimodal data.

Overall, the authors demonstrated a good effort to strengthen the connection between the theoretical insights and real-world LLM applications through the rebuttal process.

---

### Decision · Program_Chairs · 2025-01-22

Accept (Poster)